# Are We Measuring Oversmoothing in Graph Neural Networks Correctly?

**Kaicheng Zhang**[*]
School of Mathematics and Maxwell Institute
University of Edinburgh
`K.Zhang-60@sms.ed.ac.uk`

**Piero Deidda**[*]
Gran Sasso Science Institute
Scuola Normale Superiore
`piero.deidda@sns.it`

**Desmond Higham**
School of Mathematics and Maxwell Institute
University of Edinburgh
`d.j.higham@ed.ac.uk`

**Francesco Tudisco**
School of Mathematics and Maxwell Institute
University of Edinburgh
Gran Sasso Science Institute
Miniml.AI
`f.tudisco@ed.ac.uk`

## Abstract

Oversmoothing is a fundamental challenge in graph neural networks (GNNs): as the number of layers increases, node embeddings become increasingly similar, and model performance drops sharply. Traditionally, oversmoothing has been quantified using metrics that measure the similarity of neighbouring node features, such as the Dirichlet energy. We argue that these metrics have critical limitations and fail to reliably capture oversmoothing in realistic scenarios. For instance, they provide meaningful insights only for very deep networks, while typical GNNs show a performance drop already with as few as 10 layers. As an alternative, we propose measuring oversmoothing by examining the numerical or effective rank of the feature representations. We provide extensive numerical evaluation across diverse graph architectures and datasets to show that rank-based metrics consistently capture oversmoothing, whereas energy-based metrics often fail. Notably, we reveal that drops in the rank align closely with performance degradation, even in scenarios where energy metrics remain unchanged. Along with the experimental evaluation, we provide theoretical support for this approach, clarifying why Dirichlet-like measures may fail to capture performance drop and proving that the numerical rank of feature representations collapses to one for a broad family of GNN architectures.

## 1 Introduction

Graph neural networks (GNNs) have emerged as a powerful framework for learning representations from graph-structured data, with applications spanning knowledge retrieval and reasoning (Tian et al., 2022; Peng et al., 2023), personalised recommendation systems (Peng et al., 2022; Damianou et al., 2024), social network analysis (Fan et al., 2019), and 3D mesh classification (Shi & Rajkumar, 2020). Central to most GNN architectures is the message-passing paradigm, where node features are iteratively aggregated from their neighbours and transformed using learned functions, such as multi-layer perceptrons or graph-attention mechanisms.

However, the performance of message-passing-based GNNs is known to deteriorate after only a few layers, essentially placing a limit on their depth. This issue, often linked to the increasingly similar learned features as GNNs deepen, is known as oversmoothing (Li et al., 2018; Nt & Maehara, 2019; Wu et al., 2022; Rusch et al., 2023a; Zhao et al., 2024; Arnaiz-Rodríguez & Errica, 2025).

In recent years, oversmoothing in GNNs, as well as methods to alleviate it, have been studied based on the decay of some node feature similarity metrics, such as the Dirichlet energy and its variants (Oono & Suzuki, 2019; Cai & Wang, 2020; Bodnar et al., 2022; Nguyen et al., 2022; Di Giovanni

---

[*]Equal Contribution

et al., 2023; Wu et al., 2023; Roth & Liebig, 2023). At a high level, most of these metrics directly measure the norm of the absolute deviation from the dominant eigenspace of the message-passing matrix. In linear GNNs without bias terms, this eigenspace is often known and easily computable via e.g. the power method. However, when nonlinear activation functions or biases are used, the dominant eigenspace may change, causing these oversmoothing metrics to fail and give false negative signals about the oversmoothing state of the learned features.

While these metrics are often considered as sufficient but not necessary evidence for oversmoothing (Rusch et al., 2023a), there is a considerable body of literature using these unreliable metrics as their evidence for non-occurrence of oversmoothing in GNNs (Zhou et al., 2021; Chen et al., 2022; Rusch et al., 2022; Wang et al., 2022; Maskey et al., 2023; Nguyen et al., 2023; Rusch et al., 2023b; Epping et al., 2024; Scholkemper et al., 2024; Wang & Cho, 2024; Roth, 2024; Wang et al., 2025; Arnaiz-Rodriguez & Errica, 2025).

However, as we show in Section 6, the performance degradation of GNNs trained on real datasets often happens well before any noticeable decay in these oversmoothing metrics can be observed. The vast majority of empirical studies in the literature that observe the decay of Dirichlet-like energy metrics are conducted over the layers of very deep but untrained (with randomly sampled weights) or effectively untrained[1] GNNs (Wang et al., 2022; Rusch et al., 2022; 2023b; Wu et al., 2023; Roth, 2024; Wang et al., 2025), where the decay of the metrics is only driven by the small weight initialization. Instead, we show that when GNNs of different depths are trained with proper weight initialization, these metrics do not correlate with the model's performance degradation.

Furthermore, we argue that these metrics can only indicate oversmoothing when their values converge exactly to zero, corresponding to either an exact alignment to the dominant eigenspace or to the feature representation matrix collapsing to the all-zero matrix. This double implication presents an issue: in realistic settings with a large but not excessively large number of layers, we may observe the decay of the oversmoothing metric by, say, two orders of magnitude while still being far from zero. In such cases, it is unclear whether the features are aligning with the dominant eigenspace, simply decreasing in magnitude, or exhibiting neither of the two behaviours. As a result, these types of metrics provide little to no explanation for the degradation of GNN performance.

As an alternative to address these shortcomings, we advocate for the use of a continuous approximation of the rank of the network's feature representations to measure oversmoothing. Collapse of the rank of the feature representations was already considered as a cause of oversmoothing, e.g. in (Guo et al., 2023b; Roth & Liebig, 2023), however rank rank-based measures were never explicitly studied, i.e. they were never compared with other oversmoothing measures and their capacity in measuring oversmoothing was never quantified, theoretically and/or numerically. Our work fills this gap. Indeed, our experimental evaluation across various GNN architectures trained for node classification demonstrates that continuous rank relaxations, such as the numerical rank and the effective rank, correlate strongly with performance degradation in independently trained GNNs—even in settings where popular energy-like metrics show little to no correlation.

Overall, the main contributions of this paper are as follows:

- We review popular oversmoothing metrics in the current literature and simplify their theoretical analysis from a novel perspective of nonlinear activation eigenvectors.
- We notice that the rank can be a better metric for quantifying oversmoothing, and thereby we redefine oversmoothing in GNNs as the convergence towards a low-rank matrix rather than to a matrix of exactly rank one.
- We provide extensive numerical evidence that continuous rank relaxation functions provide a much more compelling measure of oversmoothing than commonly used Dirichlet-like metrics

Additionally, we investigate theoretically the causes of decay of the numerical rank. In particular, we show that both the aggregation matrices and the nonlinear activation functions can contribute to the decay. Our theoretical study is restricted to linear GNNs and nonlinear and non-negative GNNs where the eigenvector of the message-passing matrix is also the eigenvector of the nonlinear activation function. For these models, we prove that the numerical rank of the features converges exactly to one. Such results provide theoretical support to our empirical evidence that oversmoothing may occur

---

[1]Deep networks (with, say, over 100 layers) that are trained but whose loss and accuracy remain far from acceptable.

independently of the weights' magnitude and align with our perspective on oversmoothing from the point of view of nonlinear activation functions.

## 2 BACKGROUND

### 2.1 GRAPH CONVOLUTIONAL NETWORK

Let $\mathcal{G} = (\mathcal{V}, \mathcal{E})$ be an undirected graph with $\mathcal{V}$ denoting its set of vertices and $\mathcal{E} \subseteq \mathcal{V} \times \mathcal{V}$ its set of edges. Let $\widetilde{A} \in \mathbb{R}^{N \times N}$ be the unweighted adjacency matrix with $N = |\mathcal{V}|$ being the total number of nodes, $|\mathcal{E}|$ being the total number of edges of $\mathcal{G}$ and $A$ the corresponding symmetric adjacency matrix normalized by the node degrees: $A = \widetilde{D}^{-1/2} \widetilde{A} \widetilde{D}^{-1/2}$, where $\widetilde{D} = D + I$, $D$ is the diagonal degree matrix of the graph $\mathcal{G}$, and $I$ is the identity matrix. The rows of the feature matrix $X \in \mathbb{R}^{N \times d}$ are the concatenation of the $d$-dimensional feature vectors of all nodes in the graph. At each layer $l$, the node feature update of Graph Convolutional Network (GCN) (Kipf & Welling, 2016) follows $X^{(l+1)} = \sigma(A X^{(l)} W^{(l)})$ where $\sigma$ is a nonlinear activation function, applied component-wise, and $W^{(l)}$ is a trainable weight matrix.

### 2.2 GRAPH ATTENTION NETWORK

Graph Attention Networks (GATs) (Veličković et al., 2017; Brody et al., 2021) perform graph convolution via a layer-dependent message-passing matrix $A^{(l)}$ learned through an attention mechanism $A_{ij}^{(l)} = \mathrm{softmax}_j(\sigma_a(p_1^{(l)\top} W^{(l)\top} X_{i,:} + p_2^{(l)\top} W^{(l)\top} X_{j,:}))$ where $p_i^{(l)}$ are learnable parameter vectors, $X_{i,:}, X_{j,:}$ denote the feature of the $i$-th and $j$th nodes respectively, the activation $\sigma_a$ is typically chosen to be LeakyReLU, and $\mathrm{softmax}_j$ corresponds to the row-wise normalization $\mathrm{softmax}_j(A_{ij}) = \exp(A_{ij})/\sum_{j'} \exp(A_{ij'})$. The corresponding feature update is $X^{(l+1)} = \sigma(A^{(l)} X^{(l)} W^{(l)})$.

## 3 OVERSMOOTHING

Oversmoothing can be broadly understood as an increase in similarity between node features as inputs are propagated through an increasing number of message-passing layers, leading to a noticeable decline in GNN performance. However, the precise definition of this phenomenon varies across different sources. Some works define oversmoothing more rigorously as the alignment of all feature vectors with each other. This definition is motivated by the behaviour of a linear GCN: $X^{(l+1)} = A \cdots A X^{(0)} W^{(0)} \dots W^{(l)}$. Indeed, if $\widetilde{A}$ is the adjacency matrix of a fully connected graph, $A$ will have spectral radius equal to 1 with multiplicity 1, and $A^l$ will converge toward the eigenspace spanned by the dominant eigenvector. Precisely, $A^l \to uv^\top$ as $l \to \infty$, where $Au = u$ and $A^\top v = v$, see e.g. (Tudisco et al., 2015).

As a consequence, if the product of the weight matrices $W^{(0)} \cdots W^{(l)}$ does not diverge in the limit $l \to \infty$, then the features degenerate to a matrix having rank at most one, where all the features are aligned with the dominant eigenvector $u$. Mathematically, if we assume $u$ to be such that $\|u\| = 1$, this alignment can be expressed by stating that the difference between the features and their projection onto $u$, given by $\|X^{(l)} - uu^\top X^{(l)}\|$, converges to zero.

### 3.1 EXISTING OVERSMOOTHING METRICS

Motivated by the discussion about the linear case, oversmoothing is thus typically quantified and analysed in terms of the convergence of some node similarity metrics towards zero. In particular, in most cases, it is measured exactly by the alignment of the features with the dominant eigenvector of the matrix $A$. The most prominent metric that has been used to quantify oversmoothing is the Dirichlet energy, which measures the norm of the difference between the degree-normalized neighbouring node features (Cai & Wang, 2020; Rusch et al., 2023a)

$$E_{\mathrm{Dir}}(X) = \sum_{(i,j) \in \mathcal{E}} \left\| \frac{X_{i,:}}{u_i} - \frac{X_{j,:}}{u_j} \right\|_2^2, \tag{1}$$

where $u_i$ is the $i$-th entry of the dominant eigenvector of the message-passing matrix. It thus immediately follows from our discussion on the linear setting that $E_{\mathrm{Dir}}(X^{(l)})$ converges to zero as $l \to \infty$ for a linear GCN with converging weights product $W^{(0)} \cdots W^{(l)}$. This intuition suggests that a similar behaviour may occur for "smooth-enough" nonlinearities.

In particular, in the case of a GCN, the dominant eigenvector $u$ is defined by $u_i = \sqrt{1 + d_i}$ and (Cai & Wang, 2020) have proved that, using LeakyReLU activation functions, it holds $E_{\mathrm{Dir}}(X^{(l+1)}) \leq s_l \bar{\lambda} E_{\mathrm{Dir}}(X^{(l)})$, where $s_l = \|W^{(l)}\|_2$ is the largest singular value of the weight matrix $W^{(l)}$, and $\bar{\lambda} = (1 - \min_i \lambda_i)^2$, where $\lambda_i \in (0, 2]$ varies among the nonzero eigenvalues of the normalized graph Laplacian $\widetilde{\Delta} = I - A = I - \widetilde{D}^{-\frac{1}{2}} \widetilde{A} \widetilde{D}^{-\frac{1}{2}}$.

Similarly, in the case of GATs, the matrices $A_i$ are all row stochastic, meaning that $u_i = 1$ for all $i$. In this case, it has been proved that whenever the product of the entry-wise absolute value of the weights is bounded, that is $\|\Pi_{k=1}^{\infty}|W^{(k)}|\| < \infty$, then the following variant of the Dirichlet energy decays to zero (Wu et al., 2023)

$$E_{\mathrm{Proj}}(X) = \|X - \mathcal{P}X\|_F^2 \tag{2}$$

where $\mathcal{P} = uu^\top$ is the projection matrix on the space spanned by the dominant eigenvector $u$ of the matrices $A^{(l)}$.

Note that both these metrics, $E_{\mathrm{Proj}}$ and $E_{\mathrm{Dir}}$, can be used only if the dominant eigenvector of $A^{(l)}$ is the same for all $l$; this is, for example, the case with row stochastic matrices or when $A^{(l)} = A$ for all $l$. Moreover, both these metrics essentially measure the deviation of the feature representations from the dominant eigenspace of the aggregation matrices $A^{(l)}$. So we expect them to perform very similarly in capturing oversmoothing. In particular, it is not difficult to show that they are equivalent metrics from a mathematical point of view, i.e. there exist constants $C_1, C_2 > 0$ such that $C_1 E_{\mathrm{Dir}}(X) \leq E_{\mathrm{Proj}}(X) \leq C_2 E_{\mathrm{Dir}}(X)$, see Lemma A.1.

## 3.2 A Unifying Perspective Based on the Eigenvectors of Nonlinear Activations

We present here a unifying and more general perspective of the necessary conditions to have oversmoothing in the sense that $E_{\mathrm{Proj}}$ and $E_{\mathrm{Dir}}$ decay to zero, based on the concept of eigenvectors for a nonlinear activation function. In the interest of space, longer proofs for this and the subsequent sections are moved to Appendix A.

**Definition 3.1.** We say that a vector $u \in \mathbb{R}^N \setminus \{0\}$ is an eigenvector of the (nonlinear) activation function $\sigma : \mathbb{R}^N \to \mathbb{R}^n$ if for any $t \in \mathbb{R} \setminus \{0\}$, there exists $\mu_t \in \mathbb{R}$ such that $\sigma(tu) = \mu_t u$.

With this definition, we can now provide a unifying characterization of message-passing operators $A^{(l)}$ and activation functions $\sigma$ that guarantee the convergence of the Dirichlet-like energy metrics $E_{\mathrm{Proj}}$ and $E_{\mathrm{Dir}}$ to zero for the feature representation sequence defined by $X^{(l+1)} = \sigma(A^{(l)} X^{(l)} W^{(l)})$. Specifically, Theorem 3.2 shows that this holds provided all matrices $A^{(l)}$ share a common dominant eigenvector $u$, which is also an eigenvector of $\sigma$.

**Theorem 3.2.** Let $X^{(l+1)} = \sigma(A^{(l)} X^{(l)} W^{(l)})$, $l = 1, \ldots, L$, be a GNN such that $u$ is the dominant eigenvector of $A^{(l)}$ for any $l$ and also an eigenvector of the activation $\sigma$. If $\sigma$ is $1$-Lipschitz, namely $\|\sigma(x) - \sigma(y)\| \leq \|x - y\|$ for any $x, y$, and $\lim_{L \to \infty} \prod_{l=0}^{L} \|(I - \mathcal{P})A^{(l)}\|_2 \|W^{(l)}\|_2 = 0$, then

$$E_{\mathrm{Proj}}(X^{(L)}) \to 0 \quad as \quad L \to \infty.$$

The eigenvector assumption shared by $A^{(l)}$ and $\sigma$ recurs throughout our theoretical analysis, and it aligns with existing results in the literature. For example, in the case of GCNs, the matrix $A^{(l)} = A$ is symmetric, and thus $\|I - \mathcal{P}A^{(l)}\|_2 = \lambda_2$. Therefore, when $\sigma = \mathrm{LeakyReLU}$, we obtain the result by (Cai & Wang, 2020) as convergence to zero is guaranteed if $\|W^{(l)}\|_2 \leq \lambda_2$. Note in fact that the choice $\sigma = \mathrm{LeakyReLU}$ satisfies our eigenvector assumption since $u \geq 0$ by the Perron-Frobenius theorem, and thus $\mathrm{LeakyReLU}(tu) = \alpha tu$ with $\alpha$ depending only on the sign of $t$. Similarly, in the case of GATs, the matrices $A^{(l)}$ are stochastic for all $l$, implying that $u = \mathbb{1}$ is the constant vector with $(u)_i = 1$ for all $i$. If $\sigma = \otimes \psi$ is a nonlinear activation function acting entry-wise through $\psi$, then $\sigma(t\mathbb{1}) = \psi(t)\mathbb{1}$. Therefore, Theorem 3.2 implies that if the weights are sufficiently small, the

features align independently of the activation function used. This is consistent with the results in (Wu et al., 2023). However, we note that the bounds on the weights required by Theorem 3.2 and those in (Wu et al., 2023) on the weights $W^{(l)}$ are not identical, and it is unclear which of the two is more significant. Nonetheless, in both cases, having bounded weights along with any 1-Lipschitz pointwise activation function is a sufficient condition for $E_{\text{Proj}}$ to converge to zero as the depth grows in a GAT. In addition to offering a different and unifying theoretical perspective on the results in (Cai & Wang, 2020; Wu et al., 2023), we highlight the simplicity of our eigenvector-based proof, which provides added clarity on the theoretical understanding of this phenomenon.

## 4 Energy-like Metrics: What Can Go Wrong

Energy-like metrics such as $E_{\text{Dir}}$ and $E_{\text{Proj}}$ are among the most commonly used oversmoothing metrics. However, they suffer from inherent limitations that hinder their practical usability and informational content.

One important limitation of these metrics is that they indicate oversmoothing only in the limit of infinitely many layers, when their values converge exactly to zero. Since they measure a form of absolute distance, a small but nonzero value does not provide any meaningful information. On the other hand, convergence to zero corresponds to either perfect alignment with the dominant eigenspace or the collapse of the feature representation matrix to the all-zero matrix. While the former is a symptom of oversmoothing, the latter does not necessarily imply oversmoothing. Moreover, this convergence property requires the weights to be strongly bounded. However, in most practical cases, performance degradation is observed even in relatively shallow networks, far from being infinitely deep, and with weight magnitudes arbitrarily larger than what is prescribed by (Cai & Wang, 2020; Wu et al., 2023) or Theorem 3.2. This aligns with our intuition and what occurs in the linear case. Indeed, for a linear GCN, even when the features $X^{(l)}$ grow to infinity as $l \to \infty$, one observes that $X^{(l)}$ is dominated by the dominant eigenspace of $A$, even for finite and possibly small values of $l$, depending on the spectral gap of the graph. More precisely, the following theorem holds:

**Theorem 4.1.** *Let $X^{(l+1)} = AX^{(l)}W^{(l)}$ be a linear GCN. Let $\lambda_1, \lambda_2$ be the largest and second-largest eigenvalues (in modulus) of $A$, respectively. Assume the weights $\{W^{(l)}\}_{l=1}^{\infty}$ are randomly sampled from i.i.d. random variables with distribution $\nu$ such that $\int \log^+(\|W\|)d\nu + \int \log^+(\|W^{-1}\|)d\nu < \infty$, with $\log^+(t) = \max\{\log(t), 0\}$. If $|\lambda_2/\lambda_1| < 1$, then almost surely*

$$\lim_{l \to \infty} \frac{\|(I - \mathcal{P})X^{(l)}\|_F}{\|\mathcal{P}X^{(l)}\|_F} = 0$$

*with a linear rate of convergence $|\lambda_2/\lambda_1|$.*

In particular, the theorem above implies that $X^{(l)} = \lambda_1^l \left(uv^\top + R(l)\right)$ for some $v$, with $R(l) \sim O(|\lambda_2/\lambda_1|)^l$ and thus, when the spectral gap is large $|\lambda_2/\lambda_1| \ll 1$, $X^{(l)}$ is predominantly of rank one, even for moderate values of $l$. This results in weakly expressive feature representations, independently of the magnitude of the feature weights. This phenomenon can be effectively captured by measuring the rank of $X^{(l)}$, whereas Dirichlet-like energy measures may fail to detect it, as it would, for example, be the case when $\lambda_1 > 1$, having $X^{(l)} \to \infty$.

Another important limitation of Dirichlet-like metrics is their dependence on a specific dominant eigenspace, which must either be explicitly known or computed in advance. Consequently, their applicability is strongly tied to the specific architecture of the network. In particular, the dominant eigenvector $u$ of $A^{(l)}$ must be known and remain the same for all $l$. This requirement excludes their use in cases where $A^{(l)}$ varies with $l$.

## 5 The Rank as a Measure of Oversmoothing

Inspired by the behaviour observed in the linear case, we argue that measuring the rank of feature representations provides a more effective way to quantify oversmoothing, in alignment with recent work on oversmoothing (Guo et al., 2023b; Roth & Liebig, 2023). However, since the rank of a matrix is defined as the number of nonzero singular values, it is a discrete function and thus not

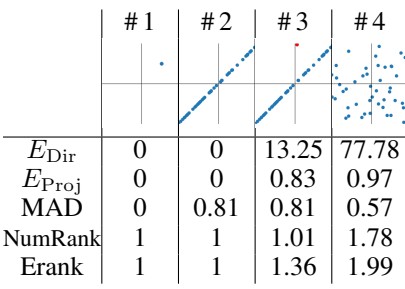

| | # 1 | # 2 | # 3 | # 4 |
|---|---|---|---|---|
| $E_{\text{Dir}}$ | 0 | 0 | 13.25 | 77.78 |
| $E_{\text{Proj}}$ | 0 | 0 | 0.83 | 0.97 |
| MAD | 0 | 0.81 | 0.81 | 0.57 |
| NumRank | 1 | 1 | 1.01 | 1.78 |
| Erank | 1 | 1 | 1.36 | 1.99 |

Figure 1: Toy scenarios depicting the behaviour of over-smoothing metrics. Each plot contains 50 nodes, each with two features plotted on the x-y axis. The features are: **# 1** of the same value; **# 2** perfectly aligned with the same vector; **# 3** aligned to the same vector except for one (red) point; **# 4** sampled from a uniform distribution. MAD (Sec. 6) and $E_{\text{Dir}}$ give false negative signals in # 3 although features are oversmoothing by definition. $E_{\text{Proj}}$ can hardly differentiate between # 3 and # 4, and is thus not robust in quantifying oversmoothing. $E_{\text{Proj}}$ and $E_{\text{Dir}}$ where computed using the first feature in place of $u$ in (1) and (2).

suitable as a measure. A viable alternative is to use a continuous relaxation that closely approximates the rank itself.

Examples of possible continuous approximations of the rank include the numerical rank, the stable rank, and the effective rank (Roy & Vetterli, 2007; Rudelson & Vershynin, 2006; Arora et al., 2019), defined as follows

$$\text{StabRank}(X) = \frac{\|X\|_*^2}{\|X\|_F^2}, \quad \text{NumRank}(X) = \frac{\|X\|_F^2}{\|X\|_2^2}, \quad \text{Erank}(X) = \exp\left(-\sum_k p_k \log p_k\right)$$

where $\|X\|_* = \sum_i \sigma_i$ is the nuclear norm, and given the singular values $\sigma_1 > \sigma_2 > \cdots > \sigma_{\min\{N,d\}}$ of $X$, $p_k = \sigma_k / \sum_i \sigma_i$ is defined as the $k$-th normalized singular value. These rank relaxation measures exhibit similar empirical behaviour as shown in Section 6.

In practice, measuring oversmoothing in terms of a continuous approximation of the rank helps to address the limitations of Dirichlet-like measures. Specifically, it offers the following advantages: (a) it is scale-invariant, meaning it remains informative even when the feature matrix converges to zero or explodes to infinity; (b) it does not rely on a fixed, predetermined eigenspace but instead captures convergence of the feature matrix toward an arbitrary lower-dimensional subspace; (c) it allows for the detection of oversmoothing in shallow networks without requiring exact convergence to rank one. A small value of the effective rank directly implies that the feature representations are low-rank, suggesting a potentially suboptimal network architecture.

In Figure 1, we present a toy example illustrating that classical oversmoothing metrics fail to correctly capture oversmoothing unless the features are perfectly aligned. This observation implies that these metrics can quantify oversmoothing only when the rank of the feature matrix converges exactly to one. In contrast, continuous rank functions provide a more reliable measure of approximate feature alignment. Later, in Figure 2, we demonstrate that the same phenomenon occurs in GNNs trained on real datasets, where exact feature alignment is rare. In such cases, classical metrics remain roughly constant, whereas the rank decreases, coinciding with a sharp drop in GNN accuracy.

### 5.1 THEORETICAL ANALYSIS OF RANK DECAY

In this section, we provide an analytical study proving the decrease of the numerical rank for a broad class of graph neural network architectures under the assumption of linear models or nonlinear models with weight matrices that are entry-wise nonnegative. Our results rigorously show that in these settings, oversmoothing can occur independently of the weight (and thus feature) magnitude and shed light on the possible causes of rank decay.

We begin with several useful observations. Let $u$ be the dominant eigenvector of $A$ corresponding to $\lambda_1$ and satisfying $\|u\| = 1$. Consider the projection $\mathcal{P} = uu^\top$. Given a matrix $X$, we can decompose it as $X = \mathcal{P}X + (I - \mathcal{P})X$. Since $u$ is a unit vector, it follows that $\|\mathcal{P}\|_2 = 1$, and therefore,

$$\|X\|_2 = \|\mathcal{P}\|_2 \|X\|_2 \geq \|\mathcal{P}X\|_2. \tag{3}$$

Moreover, since $\mathcal{P}X$ and $(I - \mathcal{P})X$ are orthogonal with respect to the Frobenius inner product, we have $\|\mathcal{P}X + (I - \mathcal{P})X\|_F^2 = \|\mathcal{P}X\|_F^2 + \|(I - \mathcal{P})X\|_F^2$. Thus, we obtain the following bound:

$$\text{NumRank}(X) = \frac{\|\mathcal{P}X + (I - \mathcal{P})X\|_F^2}{\|X\|_2^2} = \frac{\|\mathcal{P}X\|_F^2 + \|(I - \mathcal{P})X\|_F^2}{\|X\|_2^2} \leq 1 + \frac{\|(I - \mathcal{P})X\|_F^2}{\|X\|_2^2}. \tag{4}$$

The above inequality, together with Theorem 4.1, allows us to establish the convergence of the numerical rank for linear networks.

**The Linear Case** Consider a linear GCN of the form $X^{(l+1)} = AX^{(l)}W^{(l)}$, where $A$ has a simple dominant eigenvalue $\lambda_1$ satisfying $|\lambda_1| \geq |\lambda_2|$. We have already noted that $\|X\|_2 \geq \|\mathcal{P}X\|_2$, meaning that the numerical rank converges to one if $\|(I - \mathcal{P})X\|_F/\|X\|_2$ decays to zero. This occurs whenever the features grow faster in the direction of the dominant eigenvector than in any other direction. As established in Theorem 4.1, this is almost surely the case in linear GNNs. As a direct consequence, we obtain the following result:

**Theorem 5.1.** *Let $X^{(l+1)} = AX^{(l)}W^{(l)}$ be a linear GCN. Under the same assumptions as in Theorem 4.1, the following identity holds almost surely:*

$$\lim_{l \to \infty} \mathrm{NumRank}(X^{(l)}) = 1.$$

Extending the result above to general GNNs with nonlinear activation functions is highly nontrivial. However, a simplified setting to study is the one where the weights are nonnegative. Indeed, exactly as in the linear case, while generally the rank decreases only on average, considering nonnegative weights yields a monotone decrease.

**The Nonnegative Nonlinear Case** To study the case of networks with nonlinear activations, we make use of tools from the nonlinear Perron-Frobenius theory; we refer to (Lemmens & Nussbaum, 2012; Gautier et al., 2023) and the reference therein for further details.

We assume all the intermediate features of the network to be in the positive open cone $\mathcal{K} := \mathbb{R}_+^N = \{x \in \mathbb{R}^N \mid x_i > 0 \ \forall i = 1, \ldots, N\}$. Here, we consider the partial ordering $x \leq_\mathcal{K} y$ ($x \ll_\mathcal{K} y$) if and only if $y - x \geq 0$ ($y - x > 0$), where the inequalities have to be understood entrywise. Given two points $x, y \in \mathcal{K}$, let

$$d_H(x, y) = \log\left(\max_i \frac{x_i}{y_i} \cdot \max_i \frac{y_i}{x_i}\right) \tag{5}$$

denote the Hilbert distance. Note that $d_H$ is not a distance on $\mathcal{K}$, indeed $d_H(\alpha x, \beta y) = d_H(x, y)$ for any $x, y \in \mathcal{K}$ and $\alpha, \beta > 0$. However, it is a distance up to scaling; that is, it becomes a distance whenever we restrict ourselves to a slice of the cone. Because of this property, $d_H$ is particularly useful for studying the behavior of the rank of the features, which is a scale-invariant function. Indeed, the next result shows that, under mild assumptions, nonnegative weights generate layers that are nonexpansive in Hilbert distance.

**Lemma 5.2.** *Let $A$ be a nonnegative and irreducible matrix with dominant eigenvector $u \in \mathcal{K}$. Assume $X$ to be strictly positive, $W$ nonnegative with $\min_j \max_i W_{ij} > 0$, and $\sigma$ a continuous (nonlinear) function that is order preserving, subhomogeneous, and such that $u$ is also an eigenvector of $\sigma$. Then*

$$\max_i d_H\left(\sigma((AXW)_{:,i}), u\right) \leq \beta \max_i d_H(X_{:,i}, u), \tag{6}$$

*with $0 \leq \beta \leq 1$. Where $Y_{:,i}$ denotes the $i$-th column of $Y$. In particular, if $A$ is contractive in the Hilbert distance or $\sigma$ is strictly sub-homogeneous, then $\beta < 1$.*

In the above statement, order-preserving means that given any $x, y \in \mathcal{K}$ with $x \geq_\mathcal{K} y$, it holds $\sigma(x) \geq_\mathcal{K} \sigma(y)$, while (strictly) subhomogenenous means that $\sigma(\lambda x)(\ll_\mathcal{K}) \leq_\mathcal{K} \lambda\sigma(x)$ for all $x \in \mathcal{K}$ and any $\lambda > 1$. We recall that, as discussed in (Sittoni & Tudisco, 2024; Piotrowski et al., 2024), a broad range of activation functions commonly used in deep learning is subhomogeneous and order-preserving on $\mathcal{K}$. In particular, whenever an activation function is monotone increasing on $\mathbb{R}_+$, it is trivially also order-preserving. Additionally, as we prove in Proposition A.6, if the activation function is homogeneous, e.g. LeakyReLU, then any nonnegative vector is an eigenvector of $\sigma$. By contrast, if $\sigma$ is strictly subhomogeneous, e.g. $\tanh$, then the only strictly positive eigenvector is the entrywise constant one.

Next, we prove that for neural networks with nonnegative feature representations, the numerical rank goes to 1 as the depth grows to infinity.

**Theorem 5.3.** *Consider a GNN of the form $X^{(l+1)} = \sigma(A^{(l)}X^{(l)}W^{(l)})$ with $X_{:,i}^{(l)} \in \mathcal{K}$ for any $i = 1, \ldots, d$. If there exists $u \in \mathcal{K}$ such that $\lim_{l \to \infty} \max_i d_H(X_{:,i}^{(l)}, u) = 0$, then*

$$\lim_{l \to \infty} \mathrm{NumRank}(X^{(l)}) = 1.$$

| Dataset | Model | $E_{\text{Dir}}$ | | $E_{\text{Proj}}$ | | MAD | Erank | NumRank | Accuracy ratio |
|---------|-------|----------|------------|----------|------------|-----|-------|---------|----------------|
| | | Standard | Normalized | Standard | Normalized | | | | |
| Cora | GCN | -0.7871 | 0.6644 | -0.8106 | -0.8309 | -0.2460 | **0.9724** | 0.5885 | 0.2693 |
| | GAT | -0.9189 | 0.6703 | -0.9469 | -0.6054 | 0.8251 | **0.9722** | 0.7612 | 0.2493 |
| Citeseer | GCN | -0.8442 | 0.4350 | -0.8913 | -0.8667 | -0.7169 | **0.9700** | 0.6795 | 0.4380 |
| | GAT | -0.9576 | 0.0664 | -0.9585 | -0.9080 | 0.3722 | **0.9915** | 0.8047 | 0.4672 |
| Pubmed | GCN | -0.9068 | 0.7006 | -0.8508 | -0.1109 | 0.6205 | **0.9464** | 0.9268 | 0.5225 |
| | GAT | -0.8735 | -0.3684 | -0.8541 | -0.4102 | -0.3932 | 0.9270 | **0.9721** | 0.5564 |
| Squirrel | GCN | -0.7774 | 0.4171 | -0.7602 | -0.3258 | -0.8247 | 0.6316 | **0.9582** | 0.8457 |
| | GAT | -0.6864 | -0.5503 | -0.7364 | -0.7253 | 0.5002 | **0.8538** | 0.6840 | 0.7533 |
| Chameleon | GCN | -0.9223 | 0.1504 | -0.9163 | -0.8201 | -0.8809 | **0.9387** | 0.9014 | 0.6195 |
| | GAT | -0.8721 | 0.1942 | -0.9089 | -0.8234 | 0.2803 | **0.9446** | 0.8799 | 0.6332 |
| Amazon Ratings | GCN | -0.9297 | 0.8809 | -0.9079 | -0.3423 | 0.9201 | **0.9301** | 0.8049 | 0.8562 |
| | GAT | -0.9388 | 0.5277 | -0.9089 | -0.1617 | 0.6545 | **0.9248** | 0.8764 | 0.8384 |
| OGB-Arxiv | GCN | 0.7738 | 0.9194 | 0.5740 | -0.2738 | 0.2822 | **0.9682** | 0.9091 | 0.0939 |
| | GAT | -0.4097 | 0.9439 | -0.7230 | 0.8985 | 0.8492 | 0.7740 | **0.9781** | 0.2310 |
| Average correlation | | -0.7179 | 0.4036 | -0.7571 | -0.4504 | 0.1601 | **0.9103** | 0.8374 | |

Table 1: Correlation between the classification accuracy and the logarithm of metric values on GNNs with LeakyReLU and depths ranging from 2 to 24 layers, separately trained on different homophilic (Cora, Citeseer, Pubmed), heterophilic (Squirrel, Chameleon, Amazon Ratings) and large-scale (OGB-Arxiv) datasets. For Erank and NumRank, we subtract 1 so that both metrics approach zero. The rightmost column reports the ratio of classification accuracy between GNNs with 2 and 24 layers. Some heterophilic datasets may be more resilient to the increasing network depth, in-line with observations from the literature, e.g. (Guo et al., 2023a). Additional results on other datasets, activation functions and additional network components are presented in Appendix E and F.

Theorem 5.3 requires that the Hilbert distance between the feature representation and a fixed vector $u$ goes to zero. Note that this is implied by the relative metrics $E_{\text{Dir}}(X)/\|X\|$ or $E_{\text{Proj}}(X)/\|X\|$ going to zero, but is actually quite weaker. Note also that the bound (6) in Lemma 5.2 directly provides guidance on situations where the hypotheses of Theorem 5.3 are satisfied. We discuss several such situations along with some alternative and possibly weaker assumptions for Theorem 5.3 in detail in Appendix A.6. Finally, we recall that, because of Lemma 5.2, the last result applies either to GCNs with homogeneous activation function or GATs with any kind of activation function. The convergence of the numerical rank to 1 may not hold, as discussed in Appendix D.

## 6 EXPERIMENTS

The vast majority of empirical studies in the literature that observe the decay of Dirichlet-like energy metrics are conducted over the layers of deep but untrained networks (Wang et al., 2022; Rusch et al., 2022; 2023b; Wu et al., 2023; Roth, 2024; Wang et al., 2025). Moreover, the measurements are done by looking at the layers of a single network, rather than different networks of increasing depth. We emphasize that this is an overly simplified and unrealistic setting. In this section, we perform an extensive numerical investigation on the behaviour of different oversmoothing metrics when measured on networks of different depths $l = 2, \ldots, L$, trained in isolation at different depths. Our analysis shows that GNN suffer from significant performance degradation after only few-layers, at which stage the convergence patterns of Dirichlet-like metrics are difficult to observe, while relaxed rank metric already show a significant decrease, well-correlating with the performance drop.

In particular, we compare how different oversmoothing metrics behave compared to the classification accuracy, varying the GNN architectures for node classification on real-world graph data. In our experiments, we consider the following metrics:

- The Dirichlet Energy $E_{\text{Dir}}$ (Cai & Wang, 2020; Rusch et al., 2023a) and its variant $E_{\text{Proj}}$ (Wu et al., 2023). Both are discussed in Section 3.1, see in particular (1) and (2).

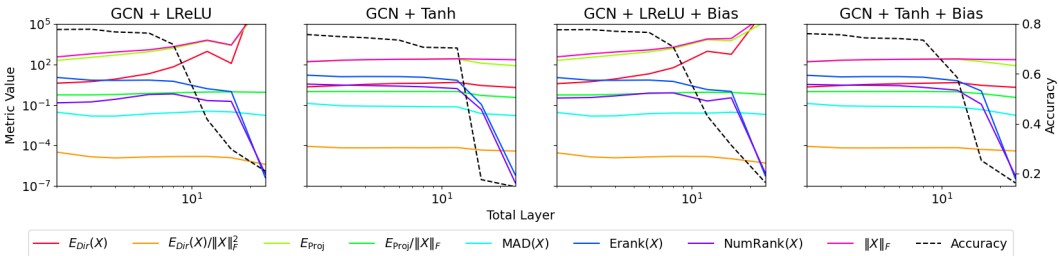

Figure 2: Four examples of the metric behaviours computed at the last hidden layer of separately trained GCNs of increasing depths. For Erank and Numrank, we measure $\mathrm{Erank}(X) - r_{\mathrm{ER}}^*$ and $\mathrm{NumRank}(X) - r_{\mathrm{NR}}^*$ for some $r^* > 1$. In these particular cases, $r_{\mathrm{ER}}^* < 1.85$, $r_{\mathrm{NR}}^* < 1.3$. Note that the effective rank and numerical rank of the input features $X^{(0)}$ are about 1084 and 13.6, respectively. Additional results are attached in Appendix G.

- Normalized versions of Dirichlet energy and its variant, $E_{\mathrm{Dir}}(X)/\|X\|_F^2$ and $E_{\mathrm{Proj}}(X)/\|X\|_F$. Indeed, from our previous discussion, a robust oversmoothing measure should be scale invariant with respect to the features. Metrics with global normalization like the ones we consider here have also been proposed in (Di Giovanni et al., 2023; Roth & Liebig, 2023; Maskey et al., 2023).
- The Mean Average Distance (MAD) (Chen et al., 2020)

$$\mathrm{MAD}(X) = \tfrac{1}{|\mathcal{E}|} \sum_{(i,j)\in\mathcal{E}} \left( 1 - \tfrac{X_{i,:}^\top X_{j,:}}{|X_{i,:}||X_{j,:}|} \right).$$

It measures the cosine similarity between the neighbouring nodes. Unlike previous baselines, this oversmoothing metric does not take into account the dominant eigenvector of the matrices $A^{(l)}$.
- Relaxed rank metrics: We consider the Numerical Rank and Effective Rank. Both are discussed in Section 5. We point out that from our theoretical investigation, in particular from (4), the numerical rank decays to 1 faster than the decay of the normalized $E_{\mathrm{Proj}}$ energy to zero. This further supports the use of the Numerical Rank as an improved measure of oversmoothing.

In Table 1 and Figure 2, we train GNNs of a fixed hidden dimension equal to 32 on homophilic, heterophilic and large-scale datasets in their default splits. We follow the standard setups of GCN and GAT as stated in Sections 2.1 and 2.2, and use homogeneous LeakyReLU (LReLU) as the activation function. For each configuration, GNNs of eight different depths ranging from 2 to 24 are trained. The oversmoothing metric and accuracy results are averaged over 10 separately trained GNNs. All GNNs are trained with NAdam Optimizer and a constant learning rate of 0.01. The oversmoothing metrics are computed at the last hidden layer before the output layer. In Figure 2 and in Appendix G, we plot the behaviour of the different oversmoothing measures, the norm of the features, and the accuracy of the trained GNNs with increasing depth. These figures clearly show that the network suffers a significant drop in accuracy, which is not matched by any visible change in standard oversmoothing metrics. By contrast, the rank of the feature representations decreases drastically, following closely the behaviour of the network's accuracy. These findings are further supported by the results shown in Table 1, where we compute the Pearson correlation coefficient between the logarithm of every measure and the classification accuracy of every GNN model. The use of a logarithmic transformation is based on the understanding that oversmoothing grows exponentially with the length of the network.

Extensions of these results are provided in Appendices E and F. In Appendix D, we perform an asymptotic ablation study on very deep (300-layer) synthetic networks with randomly sampled, untrained weights. This study serves to validate our theoretical findings on the convergence of relaxed rank metrics and to demonstrate that such untrained settings offer little insight into the ability of existing metrics to quantify oversmoothing in realistic, trained networks.

## 7 CONCLUSION

In this paper, we have discussed the problem of quantifying oversmoothing in message-passing GNNs. After simplifying the existing theoretical analysis using nonlinear activation eigenvectors and discussing the limitations of the leading oversmoothing measures, we propose the use of the rank of the features as a better measure of oversmoothing. We provide extensive experiments to

validate the robustness of the effective rank against the classical measures. In addition, we have analysed theoretically the decay of the rank of the features for message-passing GNNs. Notably, our work opens new possible research directions that investigate the use of rank-based metrics in novel architectures that can alleviate the oversmoothing phenomenon. Indeed, Dirichlet-like metrics have been employed not only to measure oversmoothing but also to design new architectures able to mitigate this phenomenon, e.g. by considering residual architectures where the magnitude of the residual is regulated according to the Dirichlet energy of neighboring nodes (Rusch et al., 2023b; Jin & Zhu, 2024). In light of our results and the superiority of rank-based metrics in capturing oversmoothing, it is natural to wonder whether the use of rank-based metrics in place of Dirichlet-like ones could better mitigate oversmoothing in similar kind of architectures. We consider this as a promising future research line.

## 8 ACKNOWLEDGEMENT

KZ was supported by the EPSRC Centre for Doctoral Training in Mathematical Modelling, Analysis and Computation (MAC-MIGS) funded by the UK Engineering and Physical Sciences Research Council (grant EP/S023291/1), Heriot-Watt University and the University of Edinburgh. PD and FT are members of the Gruppo Nazionale Calcolo Scientifico - Istituto Nazionale di Alta Matematica (GNCS-INdAM). PD was supported by the INdAM-GNCS project "NLA4ML—Numerical Linear Algebra Techniques for Machine Learning". FT is partially funded by the PRIN-MUR project MOLE (code 2022ZK5ME7) and the PRIN-PNRR project FIN4GEO (code P2022BNB97). DJH was supported by the Advanced Grant "Numerical Analysis for Stable AI" 101198795 from the European Research Council.

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

# A  PROOFS OF THE MAIN RESULTS

## A.1  EQUIVALENCE OF $E_{\text{Proj}}$ AND $E_{\text{Dir}}$

**Lemma A.1.** *Assume the graph $\mathcal{G}$ to be connected and the eigenvector $u$ to be strictly positive $u_i > 0$ for all $i$ and such that $\|u\|_2 = 1$. Then there exist $C_1 > 0$ and $C_2 > 0$ such that*

$$C_1 E_{\text{Dir}}(X) \leq E_{\text{Proj}}(X) \leq C_2 E_{\text{Dir}}(X) \qquad \forall X \in \mathbb{R}^{N \times d}.$$

*Proof.* First we show that the Dirichlet energy is equivalent to the modified Dirichlet energy where all the pairs of nodes $(i, j)$ are considered

$$E_{\text{Dir}}(X) = \sum_{(i,j)\in\mathcal{E}} \left\| \frac{X_{i,:}}{u_i} - \frac{X_{j,:}}{u_j} \right\|_2^2 \leq \sum_{i\in\mathcal{V}}\sum_{j\in\mathcal{V}} \left\| \frac{X_{i,:}}{u_i} - \frac{X_{j,:}}{u_j} \right\|_2^2 = \widetilde{E_{\text{Dir}}}(X). \tag{7}$$

Second observe that if $i, j \in \mathcal{V}$, since the graph is connected there exists some path $i_1 = 1, i_2, \ldots, i_{n+1} = j$ such that $(i_k, i_{k+1}) \in \mathcal{E}$ for all $k$. Thus, using the traingular inequality we get

$$\left\| \frac{X_{i,:}}{u_i} - \frac{X_{j,:}}{u_j} \right\|_2^2 \leq n \sum_{k=1}^{n} \left\| \frac{X_{i_k,:}}{u_{i_k}} - \frac{X_{i_{k+1},:}}{u_{i_{k+1}}} \right\|_2^2 \leq n E_{\text{Dir}}(X). \tag{8}$$

repeating the same argument for any pair of nodes $(i, j)$ we observe that for some constant $C > 1$

$$\widetilde{E_{\text{Dir}}}(X) \leq \tilde{C} E_{\text{Dir}}(X). \tag{9}$$

So, to prove the equivalence of $E_{\text{Dir}}$ and $E_{\text{Proj}}$ it is sufficient to prove the equivalence between $E_{\text{Proj}}$ and $\widetilde{E_{\text{Dir}}}$.

If we make explicit the expression of the norms in $\widetilde{E_{\text{Dir}}}$ we get

$$\widetilde{E_{\text{Dir}}}(X) = \sum_{i\in\mathcal{V}}\sum_{j\in\mathcal{V}}\sum_{k=1}^{d} \left| \frac{X_{i,k}}{u_i} - \frac{X_{j,k}}{u_j} \right|^2 = \sum_{k=1}^{d}\sum_{i\in\mathcal{V}}\sum_{j\in\mathcal{V}} \left| \frac{X_{i,k}}{u_i} - \frac{X_{j,k}}{u_j} \pm u^T X_{:,k} \right|^2$$

$$\leq \sum_{k=1}^{d}\sum_{i\in\mathcal{V}}\sum_{j\in\mathcal{V}} \frac{2}{u_i} \left| X_{i,k} - (u^T X_{:,k})u_i \right|^2 + \frac{2}{u_j} \left| X_{j,k} - (u^T X_{:,k})u_j \right|^2 \tag{10}$$

$$\leq \frac{4|\mathcal{V}|}{\min_i\{u_i\}} \sum_{k=1}^{d} \left\| X_{:,k} - uu^T X_{:,k} \right\|^2 = \frac{4|\mathcal{V}|}{\min_i\{u_i\}} E_{\text{Proj}}(X).$$

To prove the opposite observe the following

$$E_{\text{Proj}}(X) = \sum_{k=1}^{d}\sum_{i\in\mathcal{V}} u_i \left| \frac{X_{i,k}}{u_i} - (u^T X_{:,k}) \right|^2 = \sum_{k=1}^{d}\sum_{i\in\mathcal{V}} u_i \left| \frac{X_{i,k}}{u_i} - \left( \sum_h u_h^2 \frac{X_{h,k}}{u_h} \right) \right|^2$$

$$\leq \max_j\{u_j\} \sum_{k=1}^{d}\sum_{i\in\mathcal{V}} \max_j \left| \frac{X_{i,k}}{u_i} - \frac{X_{j,k}}{u_j} \right|^2 \tag{11}$$

$$\leq \max_j\{u_j\} \sum_{k=1}^{d}\sum_{i\in\mathcal{V}}\sum_{j\in\mathcal{V}} \left| \frac{X_{i,k}}{u_i} - \frac{X_{j,k}}{u_j} \right|^2 = \widetilde{E_{\text{Dir}}}(X)$$

where in the first inequality we have used that, since $\|u\|_2 = 1$, $\sum_h u_h^2 \frac{X_{h,k}}{u_h}$ is a convex combination of $\left\{ \frac{X_{h,k}}{u_h} \right\}_h$. In particular, the last inequality concludes the proof.

$\square$

## A.2 Proof of Theorem 3.2

We start proving that

$$\|(I - \mathcal{P})X^{(l+1)}\|_F \le \|(I - \mathcal{P})A^{(l)}X^{(l)}W^{(l)}\|_F, \tag{12}$$

where $\mathcal{P} = uu^\top / \|u\|^2$ is the projection matrix on the linear space spanned by $u$.

To this end, let $\pi := \mathrm{span}\{uv^\top \mid v \in \mathbb{R}^d\}$ be the 1-dimensional matrix subspace of the rank-1 matrices having columns aligned to $u$. Then it is easy to note that given some matrix $X$, $(I - \mathcal{P})X$ provides the projection of the matrix $X$ on the subspace $\pi$, i.e.

$$(I - \mathcal{P})X = \mathrm{proj}_\pi(X). \tag{13}$$

Indeed $\langle (I - \mathcal{P})X, uv^\top \rangle_F = \mathrm{Tr}(vu^\top(I - uu^\top/\|u\|^2)X) = 0$. In particular, since the projection realizes the minimal distance, we have that

$$\|X - \mathcal{P}X\|_F \le \|X - uv^\top\|_F \qquad \forall v \in \mathbb{R}^d. \tag{14}$$

Now observe that $\sigma(uu^\top A^{(l-1)}X^{(l-1)}W^{(l-1)}) = u\bar{v}^\top$ for some $\bar{v}$. Indeed, writing $v^\top = u^\top A^{(l-1)}X^{(l-1)}W^{(l-1)}$, we have that the $i$-th column of $\sigma(uu^\top A^{(l-1)}X^{(l-1)}W^{(l-1)})$ is equal to $\sigma(v_i u) = \bar{v}_i u$ for some $\bar{v}_i$, because $u$ is an eigenvector of $\sigma$. As a consequence we have

$$\begin{aligned}
\|(I - \mathcal{P})X^{(l)}\|_F &\le \|X^{(l)} - \sigma(uu^\top A^{(l-1)}X^{(l-1)}W^{(l-1)})\|_F \\
&= \|\sigma(A^{(l-1)}X^{(l-1)}W^{(l-1)}) - \sigma(uu^\top A^{(l-1)}X^{(l-1)}W^{(l-1)})\|_F \\
&\le \|(I - \mathcal{P})A^{(l-1)}X^{(l-1)}W^{(l-1)}\|_F
\end{aligned} \tag{15}$$

where we have used the 1-Lipschitz property of $\sigma$. This concludes the proof of (12)

To conclude the proof of the theorem observe that, in the decomposition

$$(I - \mathcal{P})A^{(l)} = (I - \mathcal{P})A^{(l)}\mathcal{P} + (I - \mathcal{P})A^{(l)}(I - \mathcal{P}),$$

the matrix $(I - \mathcal{P})A^{(l)}\mathcal{P}$ is zero because $A^{(l)}u = \lambda_1^l u$ for any $l$. Thus

$$(I - \mathcal{P})A^{(l)} = (I - \mathcal{P})A^{(l)}(I - \mathcal{P}),$$

and, from (12) and the inequality $\|AB\|_F \le \|A\|_2\|B\|_F$, we have

$$\|(I - \mathcal{P})X^{(L)}\|_F \le \left(\Pi_{l=0}^{L-1}\|(I - \mathcal{P})A^{(l)}\|_2\|W^{(l)}\|_2\right)\|X^{(0)}\|_F.$$

So the thesis follows from the hypothesis about the product $\Pi_{l=0}^{L-1}\|(I - \mathcal{P})A^{(l)}\|_2\|W^{(l)}\|_2$.

## A.3 Proof for Theorem 4.1

Start by studying the norm of $(I - \mathcal{P})X^{(l)}$. Then looking at the shape of the powers of the Jordan blocks matrix it is not difficult to note that $\widetilde{T}^l = O(\binom{l}{N}\lambda_2^{l-N})$ for $l$ larger than $N$. In particular if we look at the explicit expression of $(I - \mathcal{P})X^{(l)}$

$$(I - P)X^{(l)} = \begin{pmatrix} 0 & (I - \mathcal{P})\widetilde{M} \end{pmatrix} \begin{pmatrix} 0 & 0 \\ 0 & \widetilde{T}^l \end{pmatrix} M^{-1}X^{(0)}W^{(0)}\ldots W^{(l-1)}, \tag{16}$$

we derive the upper bound

$$\|(I - P)X^{(l)}\|_F \le C\binom{l}{N}|\lambda_2|^{l-N}\|X^{(0)}W^{(0)}\ldots W^{(l-1)}\|_F, \tag{17}$$

for some positive constant $C$ that is independent on $l$.

Similarly we can observe that

$$\|\mathcal{P}X^{(l)}\|_F \ge \|u^\top A^l X^{(0)}W^{(0)}\ldots W^{(l-1)}\|_F =$$

$$= \|(\lambda_1^l v_1^\top + u^\top \widetilde{M}O(\binom{l}{N}\lambda_2^{l-N})\widetilde{M}')X^{(0)}W^{(0)}\ldots W^{(l-1)}\|_F \ge$$

$$\ge |\lambda_1|^l\left(\|v_1^\top X^{(0)}W^{(0)}\ldots W^{(l-1)}\|_F - \left\|u^\top \widetilde{M}O\left(\binom{l}{N}\left(\frac{\lambda_2}{\lambda_1}\right)^l\right)\widetilde{M}'X^{(0)}W^{(0)}\ldots W^{(l-1)}\right\|_F\right) \ge$$

$$\ge |\lambda_1|^l\|v_1^\top X^{(0)}W^{(0)}\ldots W^{(l-1)}\|_F\left(1 - O\left(\binom{l}{N}\left|\frac{\lambda_2}{\lambda_1}\right|^l \frac{\|X^{(0)}W^{(0)}\ldots W^{(l-1)}\|_F}{\|v_1^\top X^{(0)}W^{(0)}\ldots W^{(l-1)}\|_F}\right)\right)$$

$$\tag{18}$$

Now observe that under the randomness hypothesis from (Furstenberg & Kifer, 1983) and more generally from the Oseledets ergodic multiplicative theorem, we have that for almost any the limit $w \in \mathbb{R}^d \lim_{l \to \infty} \frac{1}{l} \log \|w^\top W^{(0)} \ldots W^{(l-1)}\| = c(\nu)$ exists and is equal to the maximal Lyapunov exponent of the system. In particular for any $w$ and $\epsilon > 0$ there exists $l_{w,\epsilon}$ sufficiently large such that for any $l > l_{w,\epsilon}$

$$c(\nu) - \epsilon \leq \frac{1}{l} \log \|w^\top W^{(0)} \ldots W^{(l-1)}\| < c(\nu) + \epsilon \tag{19}$$

i.e.

$$e^{l(c(\nu)-\epsilon)} \leq \|w^\top W^{(0)} \ldots W^{(l-1)}\| < e^{l(c(\nu)+\epsilon)} \quad \forall l \geq l_{w,\epsilon}. \tag{20}$$

Now take as vector $w$ first the rows of $X^{(0)}$ and then the vector $v_1^\top X^{(0)}$, then almost surely for any $\epsilon$ there exists $l_\epsilon$ such that for any $l > l_\epsilon$

$$e^{l(c(\nu)-\epsilon)} \leq \|w^\top W^{(0)} \ldots W^{(l-1)}\| < e^{l(c(\nu)+\epsilon)}, \tag{21}$$

holding for any $l \geq l_\epsilon$ and any $w \in \{v_1\} \cup \{X_0^\top e_i\}_{i=1}^N$.

Next recall that $\|X^{(0)} W^{(0)} \ldots W^{(l-1)}\|_F = \sqrt{\sum_i \|e_i^\top X^{(0)} W^{(0)}\|^2}$, meaning that almost surely, for $l \geq l_\epsilon$:

$$N e^{l(c(\nu)-\epsilon)} \leq \|X^{(0)} W^{(0)} \ldots W^{(l-1)}\|_F \leq N e^{l(c(\nu)+\epsilon)}. \tag{22}$$

In particular for any $\epsilon$, there exists $l$ sufficiently large such that

$$\left( \binom{l}{N} \left| \frac{\lambda_2}{\lambda_1} \right|^l \frac{\|X^{(0)} W^{(0)} \ldots W^{(l-1)}\|_F}{\|v_1^\top X^{(0)} W^{(0)} \ldots W^{(l-1)}\|_F} \right) \leq \left( \binom{l}{N} \left| \frac{\lambda_2}{\lambda_1} \right|^l e^{2l\epsilon} \right) \tag{23}$$

and thus, since $|\lambda_2| < |\lambda_1|$ and we can choose $\epsilon$ arbitrarily small, almost surely it has limit equal to zero. In particular we can write

$$\lim_l \frac{\|(I - \mathcal{P} X^{(l)})\|_F}{\|\mathcal{P} X^{(l)}\|_F} \sim \lim_l \frac{\binom{l}{N} |\lambda_2|^{l-N} \|X^{(0)} W^{(0)} \ldots W^{(l)}\|_F}{|\lambda_1|^l \|v_1^\top X^{(0)} W^{(0)} \ldots W^{(l)}\|_F} = 0 \tag{24}$$

where we have used the same argument as before to state that the limit is zero.

### A.4 Proof of Lemma 5.2

Since $A$ is nonnegative, then from Perron Frobenius theory (Lemmens & Nussbaum, 2012), we know that

$$d_H\Big((AX)_{:,i}, u\Big) = d_H\Big((AX)_{:,i}, \lambda_1(A)u\Big) = d_H\Big((AX)_{:,i}, Au\Big) \leq \beta d_H\big(X_{:,i}, u\big) \qquad \forall i. \tag{25}$$

for some $\beta \leq 1$, where we have used $\lambda_1(A) > 0$ and the scaling invariant property of the Hilbert distance. In particular if $A$ is contractive in Hilbert distance $\beta < 1$.

Then note that, for any $i$, we can write $(AXW)_{:,j}$ as follows

$$\big(AXW\big)_{:,j} = \sum_j W_{ij}(AX)_{:,j}. \tag{26}$$

Thus we **CLAIM** that given $x_1, x_2, y \in \mathcal{K}$ then

$$d_H(x_1 + x_2, y) \leq \max\{d_H(x_1, y), d_H(x_2, y)\}. \tag{27}$$

Observe that if the claim holds, by induction it can trivially be extended from 2 to $d$ points yielding

$$d_H\Big((AXW)_{:,j}, u\Big) \leq \max_i d_H\Big(W_{ij}(AX)_{:,i}, u\Big) \leq \max_j d_H\Big((AX)_{:,j}, u\Big) \leq \beta \max_j d_H\Big(X_{:,j}, u\Big), \tag{28}$$

where we have used the scale-invariance property of the Hilbert distance and the fact that $\max_i W_{ij} > 0$ for all $j$.

We miss to prove the claim. To this end, exploiting the expression of the Hilbert distance we write

$$
\begin{aligned}
d_H(x_1 + x_2, y) &= \log\left(\sup_j \sup_i \frac{(x_1)_i + (x_2)_i}{(y)_i} \frac{(y)_j}{(x_1)_j + (x_2)_j}\right) \\
&\leq \log\left(\sup_i \sup_j \max_{x_1, x_2}\left\{\frac{(x_1)_i}{(x_1)_j}, \frac{(x_2)_i}{(x_2)_j}\right\} \frac{(y)_j}{(y)_i}\right) \\
&= \max_{x_1, x_2}\left\{d_H(x_1, y), d_H(x_2, y)\right\}.
\end{aligned}
\tag{29}
$$

concluding the proof.

Next we prove that, a continuous subhomogeneous and order-preserving function $\sigma$ with eigenvector $u$ in the cone, is not nonexpansive in Hilbert distance with respect to $u$. Formally we claim that

$$
d_H\big(\sigma(y), u\big) \leq d_H\big(y, u\big) \qquad \forall y \in \mathcal{K}.
\tag{30}
$$

To prove it, let $y \in \mathcal{K}$ and assume w.l.o.g. that $\|y\|_1 = t > 0$ and $\|u\|_1 = 1$, then

$$
M(y/tu) = \max_{i=1,\dots,N} \frac{y_i}{t(u)_i} \geq \frac{\|y\|_1}{t\|u\|} = 1 \qquad m(y/tu) = \min_{i=1,\dots,N} \frac{y_i}{t(u)_i} \leq \frac{\|y\|_1}{t\|u\|_1} = 1.
\tag{31}
$$

By definition given $x, y \in \mathcal{K}$, $m(y/x)x \leq_{\mathcal{K}} y \leq_{\mathcal{K}} M(y/x)x$. Moreover we recall that since $u$ is an eigenvector for any $t > 0$ there exists $\lambda_t > 0$ such that $\sigma(tu) = \lambda_t u$. Thus we use the subhomogeneity of $\sigma$ and the fact that $u$ is an eigenvector of $\sigma$ to get the following inequalities:

$$
m(y/tu)\lambda_t tu \leq_{\mathcal{K}} \sigma\big(m(y/tu)tu\big) \leq_{\mathcal{K}} f(y) \leq_{\mathcal{K}} f\big(M(y/tu)tu\big) \leq_{\mathcal{K}} M(y/x_c)\lambda_t tu,
\tag{32}
$$

where the inequalities are strict if $\sigma$ is strictly subhomogeneous. In particular we have $m(f(y)/tu) \geq \lambda_t m(y/tu)$ and $M(f(y)/tu) \leq \lambda_t M(y/tu)$. Finally the last inequalities and the scale invariance property of $d_H$ yield the thesis:

$$
d_H(f(y), u) = d_H(f(y), tu) = \log\left(\frac{M(f(y)/tu)}{m(f(y)/tu)}\right) \leq \log\left(\frac{M(y/tu)}{m(y/tu)}\right) = d_H(y, tu) = d_H(y, u).
\tag{33}
$$

with the inequality that is strict if $\sigma$ is strictly subhomogeneous.

Then the thesis of the Lemma follows by applying (30) to (28).

### A.5 PROOF OF THEOREM 5.3

We will prove that, as a consequence of the hypothesis $\lim_{l\to\infty} \max_i d_H(X^{(l)}_{:,i}, u) = 0$,

$$
\lim_{l\to\infty} \frac{\|(I - \mathcal{P})X^{(l)}\|_F}{\|\mathcal{P}X^{(l)}\|_F} = 0,
\tag{34}
$$

where $\mathcal{P} = uu^T/\|u\|_2^2$. Indeed (34) is equivalent to proving that the numerical rank goes to 1 as $l$ goes to $\infty$:

$$
1 \leq \text{NumRank}(X^{(l)}) \leq 1 + \frac{\|(I - \mathcal{P})X^{(l)}\|_F^2}{\|X^{(l)}\|_2^2} \leq 1 + \frac{\|(I - \mathcal{P})X^{(l)}\|_F^2}{\|\mathcal{P}X^{(l)}\|_F^2}.
\tag{35}
$$

To prove (34), we recall from Lemma 2.5.1 in (Lemmens & Nussbaum, 2012) that for any $w$ such that $u^\top w = c$

$$
\|w - \mathcal{P}w\|_u \leq \|\mathcal{P}w\|_u (e^{d_T(w, \mathcal{P}w)} - 1),
\tag{36}
$$

where $d_T(x, y) = \log(\max\{M(x/y), m(x/y)^{-1}\})$ and where since the dual cone of $\mathbb{R}^n_+$ is $\mathbb{R}^n_+$ itself, we are considering the norm induced by $u$ on the cone, i.e. $\|x\|_u = u^\top x$ for any $x$ in the cone. In practice the norm induce by $u$ $\|\cdot\|_u$ can be defines by the Minkowki functional of the set $\Omega = \text{ConvexHull}\{\{\Omega_1\} \cup \{-\Omega_1\}\}$ where $\Omega_1 = \{x \in \mathcal{K} \ u^\top x \leq 1\}$

Then since $\|\mathcal{P}w\|_u = \|w\|_u = u^\top w$, we have that $M(w/\mathcal{P}w) \geq 1$ and $m(w/\mathcal{P}w) \leq 1$. Thus $d_T(w, \mathcal{P}w) \leq d_H(w, \mathcal{P}w)$, yielding

$$
\|w - \mathcal{P}w\|_u \leq \|\mathcal{P}w\|_u (e^{d_H(w, \mathcal{P}w)} - 1).
\tag{37}
$$

From the equivalence of the norms there exists some constant $c > 0$ such that we can equivalently write

$$\|w - \mathcal{P}w\|_2 \leq C\|\mathcal{P}w\|_2(e^{d_H(w, \mathcal{P}w)} - 1). \tag{38}$$

Now recall that the squared frobenius norm of a matrix is the sum of the squared 2-norms of the its columns, so we can apply the last inequality to the matrix $X^{(l)}$ columnwise obtaining:

$$\|(I - \mathcal{P})X^{(l)}\|_F^2 \leq C\|\mathcal{P}X^{(l)}\|_F^2 \big(e^{\max_i \{d_H(X^{(l)}_{:,i}, \mathcal{P}X^{(l)}_{:,i})\}} - 1\big)^2. \tag{39}$$

the proof is concluded using the hypothesis and observing that by the scale invariance property of the Hilbert distance $d_H(X^{(l)}_{:,i}, \mathcal{P}X^{(l)}_{:,i}) = d_H(X^{(l)}_{:,i}, u)$, yielding:

$$1 \leq \big(\mathrm{NumRank}(X^{(l)})\big) \leq 1 + \frac{\|(I - \mathcal{P})X^{(l)}\|_F^2}{\|\mathcal{P}X^{(l)}\|_F^2} \leq 1 + C\big(e^{\max_i \{d_H(X^{(l)}_{:,i}, u)\}} - 1\big)^2 \tag{40}$$

and concluding the proof.

## A.6 SOME SITUATIONS WHERE $\lim_{l \to \infty} \max_i d_H(X^{(l)}_{:,i}, u) = 0$

In this section we explore three different situations where the Hilbert distance of the features from the dominant eigenvector $u$ of the thea aggregation matrices is guaranteed to converge to 1.

**1st situation**   A first situation where the Hilbert distance of the features from the dominant eigenvector $u$ converges to zero is the case of all matrices $A^{(l)}$ are contractive.

**Lemma A.2.** *Let $A^{(l)}$ be nonnegative and irreducible matrices with dominant eigenvector $u \in \mathcal{K}$. Assume also $X^{(0)}$ to be strictly positive, $W^{(l)}$ nonnegative with $\min_j \max_i W^{(l)}_{ij} > 0$ and $\sigma \in C(\mathbb{R}^N_+, \mathbb{R}^N_+)$ a nonlinear function that is order preserving, subhomogeneous and such that $u$ is also an eigenvector of $\sigma$. Any matrix $A^{(l)}$ is known to satisfy $d_H(A^{(l)}x, A^{(l)}y) \leq d_H(x, y)$ for all $x, y \in \mathcal{K}$, then if $\lim_{k \to \infty} \prod_{l=1}^k \beta_l = 0$*

$$\lim_{l \to \infty} \max_i d_H(X^{(l)}_{:,i}, u) = 0. \tag{41}$$

*Proof.* The proof is a trivial consequence of Lemma 5.2. Indeed iterating the result in the thesis we know that

$$\max_i d_H\big(\sigma(X^{(l)}_{:,i}), u\big) \leq \prod_{i=1}^{l-1} \beta_i \max_i d_H(X^{(0)}_{:,i}, u), \tag{42}$$

concluding the proof □

In particular we remind that from the Perron-Frobenius theory any strictly positive matrix $A$ is known to be contractive of a parameter $\beta < 1$, see (Lemmens & Nussbaum, 2012).

**2nd situation**   A second situation where the Hilbert distance of the features from the dominant eigenvector $u$ converges to zero is the case of a strictly subhomogeneous activation function.

**Lemma A.3.** *Let $A^{(l)}$ be nonnegative and irreducible matrices with dominant eigenvector $u \in \mathcal{K}$. Assume also $X^{(0)}$ to be strictly positive, $W^{(l)}$ nonnegative with $\min_j \max_i W^{(l)}_{ij} > 0$ and $\sigma \in C(\mathbb{R}^N_+, \mathbb{R}^N_+)$ a nonlinear function that is order preserving, strongly subhomogeneous and such that $u$ is also an eigenvector of $\sigma$. Since $\sigma$ is strongly subhomogeneous, for any $l$ there exists $\beta_l < 1$ such that $\max_i d_H(\sigma(A^{(l)}X^{(l)}W^{(l)})_{:,i}, u) \leq \beta_l d_H((A^{(l-1)}X^{(l-1)}W^{(l-1)})_{:,i}, u)$, then if $\lim_{k \to \infty} \prod_{l=1}^k \beta_l = 0$*

$$\lim_{l \to \infty} \max_i d_H(X^{(l)}_{:,i}, u) = 0. \tag{43}$$

*Proof.* The proof is again a trivial consequence of Lemma 5.2. Indeed iterating the result in the thesis we know that

$$\max_i d_H\left(\sigma\big(X^{(l)}_{:,i}\big), u\right) \le \prod_{i=1}^{l-1} \beta_i \max_i d_H(X^{(0)}_{:,i}, u), \tag{44}$$

concluding the proof. $\qquad\square$

Note that whenever we have a strongly concave activation function $\sigma$ on $\mathbb{R}_+$, e.g. tanh, then it is strongly subhomogeneous and so the result above applies.

**3rd situation**   Here we discuss a third situation, possibly weaker than the previous ones. This is essentially an adaptation of the ergodic theorem proved in (Nussbaum, 1990). Consider the cone $\mathcal{K}' = \mathbb{R}_+^{N\times d}$ and introduce the set $\Pi = \{Y \in \mathcal{K}'$ s.t. $Y_{:,i} = \alpha_i u \; \forall i = 1,\ldots,d\}$. Moreover let $\psi \in \text{Int}(\mathcal{K}')$ and define $\Sigma_\psi = \{Y \in \mathcal{K}'$ s.t. $\psi(Y) = 1\}$. Finally given $X \in \mathcal{K}'$ define

$$\pi(X) := \underset{Y \in \Sigma_\psi}{\arg\min}\, d_H(X, Y). \tag{45}$$

**Definition A.4** (Hypotheses H). We say that a GNN $X^{(l+1)} = f^{(l)}(X^{(l)}) = \sigma(A^{(l)}X^{(l)}W^{(l)})$ with $X^{(0)} \in \mathcal{K}'$ satisfies hypotheses $[H]$ if the following conditions are verified:

H1  $A^{(l)}$ is nonnegative and $\min_j \max_i W^{(l)}_{ij} > 0$ for any $l$.

H2  The function $\sigma$ is subhomogeneous and differentiable in $\mathbb{R}_+$.

H3  $u \in \mathcal{K}$ is the dominant eigenvector of all of the matrices $A^{(l)}$ and it is also an eigenvector of $\sigma$.

H4  There exists an integer $p > 0$ and a sequence of $dN \times dN$ strictly positive matrices $\{B^{(l)}\}$ such that $\forall X$ with $m\big(X^{(lp)}, \pi(X^{(lp)})\big)\pi(X^{(lp)}) \le X \le M\big(X^{(lp)}, \pi(X^{(lp)})\big)\pi(X^{(lp)})$

$$\partial_X g^{(l)}(X) \ge B^{(l)} \quad \forall l \ge 1$$

where $g^{(l)}(X) := f^{((l+1)p-1)} \circ \cdots \circ f^{(lp)}(X)$.

H5  $\forall l \ge 0$ there exists $\eta^{(l)} > 0$ s.t. $B^{(l)}\pi(X^{(lp)}) \ge \eta^{(l)} g^{(l)}\big(\pi(X^{(lp)})\big)$.

H6  $\lim_{M\to\infty} \sum_{l=0}^M \eta^{(l)} exp\big(-\Delta(B^{(l)})\big) = \infty$ where $\Delta(B) = \sup_{x,y\in\mathcal{K}'} d_H(Bx, By) < \infty$ is the projective diameter of the matrix $B$.

**Theorem A.5.** *Let* $X^{(l+1)} = f^{(l)}(X^{(l)}) = \sigma(A^{(l)}X^{(l)}W^{(l)})$, *with* $X^{(0)} \in \mathcal{K}'$, *be a GNN satisfying hypotheses* $[H]$. *Then*

$$\lim_{l\to\infty} \max_i \big(d_H(X^{(l)}_{:,i}, u)\big) = 0.$$

*Proof.* We claim that under hypotheses [H]

$$\lim_{l\to\infty} d_H(X^{(l)}, \Pi) = 0. \tag{46}$$

As a consequence of this it is easy to note that $\lim_{l\to\infty} \max_i \big(d_H(X^{(l)}_{:,i}, u)\big) = 0$. Indeed it is not difficult to check that

$$exp\Big(d_H\big(X^{(l)}, \pi\big(X^{(l)}\big)\big)\Big) = \frac{M(X^{(l)}, \pi(X^{(l)}))}{m(X^{(l)}, \pi(X^{(l)}))} = \max_{ji}\max_{hk} \frac{X^{(l)}_{ji}\pi\big(X^{(l)}\big)_{hk}}{X^{(l)}_{hk}\pi\big(X^{(l)}\big)_{ij}} \ge$$

$$\ge \max_i \max_j \max_h \frac{X^{(l)}_{ji}\pi\big(X^{(l)}\big)_{hi}}{X^{(l)}_{hi}\pi\big(X^{(l)}\big)_{ji}} = \max_i exp\Big(d_H(X^{(l)}_{:,i}, u)\Big). \tag{47}$$

where we have used that by definition $\pi\big(X^{(l)}\big)_{:,i} = \alpha_i u$ for all $i$, where necessarily $\alpha_i > 0$ for all $i$. Otherwise, since $X^{(l)} \in \mathcal{K}'$, we would have $d_H\big(X^{(l)}, \pi\big(X^{(l)}\big)\big) = \infty$, against the minimality.

Next we prove the claim. The proof is adapted for our scopes from the proof of the weak ergodic theorem 2.1 proved in (Nussbaum, 1990) for homogeneous mappings. To simplify the notation we denote by $X := X^{(lp)}$, $\pi := \pi(X^{(lp)})$, $g := g^{(l)}$, $m := m(X^{(lp)}, \pi(X^{(lp)}))$, $B := B^{(l)}$, $\eta := \eta^{(l)}$ and $M := M(X^{(lp)}, \pi(X^{(lp)}))$. Then consider $z^1(t) = (1-t)m\pi + tX$ and $z^2(t) = (1-t)X + tM\pi$. Then from hypothesis [H4] we have the following inequalities:

$$g(X) - g(m\pi) = \int_0^1 \partial_X g(z^1(t))(X - m\pi) \geq B(X - m\pi)$$
$$g(M\pi) - g(x) = \int_0^1 \partial_X g(z^2(t))(M\pi - X) \geq B(M\pi - X). \tag{48}$$

In particular, since $g$ is subhomogeneous and by definition of $\pi$ $m \leq 1$ and $M \geq 1$ we have:

$$mg(\pi) + B(X - m\pi) \leq g(m\pi) + B(X - m\pi) \leq g(X) \leq$$
$$\leq g(M\pi) - B(M\pi - X) \leq Mg(\pi) - B(M\pi - X) \tag{49}$$

Since $B(X - m\pi) + B(M\pi - X) = (M - m)B\pi$, we can use Lemma 2.2 in (Nussbaum, 1990) we know that

$$B(X - m\pi) \geq \gamma(M - m)B\pi \qquad \text{or} \qquad B(M\pi - X) \geq \gamma(M - m)B\pi, \tag{50}$$

where $\gamma = (1/2)exp(-\Delta(B))$. Without loss of generality assume that $B(X - m\pi) \geq \gamma(M-m)B\pi$ the second case can be handled analogously. Then, since $B(M\pi - X)$, from (51), we have

$$mg(\pi) + \gamma\eta(M - m)g(\pi) \leq mg(\pi) + \gamma(M - m)B\pi \leq g(X) \leq Mg(\pi). \tag{51}$$

In particular

$$d_H(g(X), g(\pi)) \leq \log(\frac{M}{m + \eta\gamma(M - m)}) = \log\left(\frac{M}{m + \eta\gamma(M - m)}\right) =$$
$$= \log(M/m)\frac{\log\left(\frac{M}{m}\frac{1}{1+\eta\gamma(M/m-1)}\right)}{\log(M/m)} \leq \tag{52}$$
$$\leq \log(M/m)(1 - \eta\gamma) = (1 - \eta\gamma)d_H(X, \pi).$$

In the last we have used the following fact: if $\xi_1(s) = \log\big(s(1 + \eta\gamma(s - 1))^{-1}\big)$ and $\xi_2(s) = \log(s)$ with $s > 1$ then since $\xi_{1,2}(1) = 0$ and $1 + \eta\gamma(s' - 1) > 1$ for $s' > 1$, for any $s > 1$ there exists some $s' > 1$ such that

$$\xi_1(s)/\xi_2(s) = \xi_1'(s')/\xi_2'(s') = (1 - \eta\gamma)(1 + \eta\gamma(s' - 1))^{-1} \leq (1 - \eta\gamma). \tag{53}$$

Now note that $d_H(X, \pi) = d_H(X, \Pi)$ by the minimality of $\pi$ and the fact that $d_H(X, \alpha Y) = d_H(X, Y)$ for any $\alpha > 0$. Second, using [H1] and [H3] it is very easy to observe that $g(\pi) \in \Pi$. So $d_H(g(X), \Pi) \leq d_H(g(X), g(\pi))$. In conclusion we have proved that

$$d_H\big(X^{(lp)}, \Pi\big) = d_H\big(g^{(l)}(X^{(l)}), \Pi\big) \leq \left(1 - \eta^{(l)}\frac{exp\big(-\Delta(B^{(l)})\big)}{2}\right) d_H(X^{(l)}, \Pi). \tag{54}$$

In particular iterating, using (47) and recalling that $\max_i d_H(X_i^{(l_1)}, u) \leq \max_i d_H(X_i^{(l_2)}, u)$ if $l_2 > l_1$ (see Lemma 5.2) we have that for any $L > lp$:

$$\max_i d_H(X_i^{(L)}, u) \leq \max_i d_H(X_i^{(lp)}, u) \leq d_H\big(X_i^{(lp)}, \Pi\big) \leq$$
$$\leq \prod_{j=0}^l \left(1 - \eta^{(j)}\frac{exp\big(-\Delta(B^{(j)})\big)}{2}\right) d_H(X^{(0)}, \Pi). \tag{55}$$

Moreover, we have that

$$\lim_{l \to \infty} \prod_{j=0}^{l} \left( 1 - \eta^{(j)} \frac{exp\left( -\Delta(B^{(j)}) \right)}{2} \right) = 0 \iff$$

$$\lim_{l \to \infty} \sum_{j=0}^{l} -\log \left( 1 - \eta^{(j)} \frac{exp\left( -\Delta(B^{(j)}) \right)}{2} \right) = \infty \iff \tag{56}$$

$$\lim_{l \to \infty} \sum_{j=0}^{l} \left( \eta^{(j)} \frac{exp\left( -\Delta(B^{(j)}) \right)}{2} \right) = \infty,$$

which concludes the proof.

$\square$

Next, we discuss briefly the hypothesis H4 which is the most technical one. In particular, such hypothesis recalls the property of a primitive matrix. Actually we note that, if the matrices $A^{(l)}$ are primitive, under few additional mild assumptions, it is possible to prove that the hypothesis H4 is always satisfied. Assume that there exists an index $p$ such that:

1. The matrices $A^{(l)}$ with $l = 0, \ldots, p-1$ satisfy $A^{(l)} \geq \alpha_1 A^*$ where $A^*$ is the unweighted adjacency matrix of the graph; i.e. $A^*_{ij} = 1$ if the edge $(i,j)$ belongs to the graph, $A^*_{ij} = 0$ otherwise. And assume that $A^*$ is primitive of index smaller then $p$, i.e. $\left( (A^*)^p \right)_{ij} > 1$ for all $i, j$

2. $\partial_x \sigma(X_{ij}) \geq \alpha_2$ for all $X$ with $X = f^{(l)} \circ \cdots \circ f^{(0)}(X^*)$ varying $l = 0, \ldots, p - 1$ and $X^*$ among the matrices that satisfy $m\left( X^{(0)}, \pi(X^{(0)}) \right) \pi(X^{(0)}) \leq X \leq M\left( X^{(0)}, \pi(X^{(0)}) \right) \pi(X^{(0)})$.

3. $\left( W^{(0)} \cdot \ldots \cdot W^{(p-1)} \right)_{ij} > \alpha_3 > 0$ for all $i, j$.

Then for any matrix $X$ satisfying $m\left( X^{(0)}, \pi(X^{(0)}) \right) \pi(X^{(0)}) \leq X \leq M\left( X^{(0)}, \pi(X^{(0)}) \right) \pi(X^{(0)})$ we have

$$\left( \partial_X g(X) \right)_{ij} \geq \alpha_1^p \alpha_2^p \alpha_3 \quad \forall ij$$

where $g(X) := f^{(p-1)} \circ \cdots \circ f^{(0)}(X)$.

Indeed, vectorizing the matrix $X$ we can write $\partial_X f^{(i)}(X) = \text{diag}\left( \sigma'\left( (W^T \otimes A) vec(X) \right) \right)(W^T \otimes A) \geq \alpha_2 (W^T \otimes A)$. Then we can apply the chain rule to the function $g$ to get

$$\left( \partial_X g(X^*) \right) \geq \alpha_2^p \left( W^{(p-1)T} \otimes A^{(p-1)} \right) \ldots \left( W^{(0)T} \otimes A^{(0)} \right). \tag{57}$$

Finally we use the properties of the matrices $W^{(i)}$ and $A^{(i)}$ and the properties of the Kronecker product to obtain

$$\left( \partial_X g(X^*) \right) \geq \alpha_2^p \left( W^{(p-1)T} \ldots W^{(0)T} \right) \otimes \left( A^{(p-1)} \ldots A^{(0)} \right) \geq \alpha_2^p \alpha_1^p \alpha_3 \mathbb{1} \tag{58}$$

where $\mathbb{1}$ is the matrix with every entry equal to 1.

## A.7 Eigenpairs of entry-wise subhomogeneous maps

We conclude with a formal investigation of the eigenpairs of activation functions that are entry-wise subhomogeneous. Let $\sigma = \otimes^N \psi$ with $\psi \in C(\mathbb{R}, \mathbb{R})$ that is subhomogeneous on $\mathbb{R}_+$. Then one can easily show that $\sigma$ is itself subhomogeneous on $\mathbb{R}_+^N$. We have the following result,

**Proposition A.6.** *Let $\sigma = \otimes^N \psi$ with $\psi \in C(\mathbb{R}_+, \mathbb{R}_+)$ be order preserving. Then: 1) If $\sigma$ is homogeneous, any positive vector is an eigenvector of $\sigma$, 2) If $\sigma$ is strictly subhomogeneous, the only eigenvector of $\sigma$ in $\mathcal{K}$ is the constant vector.*

We start from the homogenous case. Note that since $\psi$ is homogeneous we have that necessarily $f(t) = ct$ for all $t, c \geq 0$, this in particular means that every $u \in \mathbb{R}_+^N$ is an eigenvector of $\sigma$ with corresponding eigenvalue $\lambda_1 = c$.

Then we can consider the subhomogeneous case. Assume that we have $u \in \mathbb{R}_+^N$ that is an eigenvector of $\sigma$ with eigenvalue $\lambda$ and $u_i > 0$ for all $i$, then

$$\psi(u_i) = \lambda u_i \qquad \forall i = 1, \dots, N. \tag{59}$$

By strict subhomogeneity this means that necessarily $u$ is constant, indeed if $u_i > u_j > 0$ then

$$\lambda u_j = \psi(u_j) = \psi\left(u_j \frac{u_i}{u_i}\right) > \frac{u_j}{u_i}\psi(u_i) = \lambda u_j, \tag{60}$$

yielding a contradiction. In particular any constant vector $u$ in $\mathbb{R}_+^N$ is easily proved to be an eigenvector of $\sigma$ relative to the eigenvalue $\lambda = \|\sigma(u)\|_1/\|u\|_1 = \psi(u_i)/u_i$ where $u_i$ is any entry of $u$.

## B    LIMITATIONS

Although a small effective rank or numerical rank indicates oversmoothing and can be subsequently linked to the underperformance of GNNs, a large effective rank or numerical rank does not necessarily correspond to a good network performance. Prior study has suggested some degree of smoothing can be beneficial (Keriven, 2022), and as an extreme example, features sampled from a uniform distribution and with randomly assigned labels would almost surely have a large effective rank, but they cannot be classified accurately due to the lack of any underlying pattern. We note that this limitation is not specific to the two relaxed rank measures, as the same argument is directly applicable to all other oversmoothing metrics.

Consequently, as shown in Table 4, when additional components are used to (partially) alleviate oversmoothing, particularly when residual connections are used, the accuracy ratio may remain large over the layers, and all oversmoothing metrics correlate poorly with the accuracy of GNNs. This in turn suggests these oversmoothing metrics become less informative as the oversmoothing problem is mitigated or alleviated.

## C    COMPUTATIONAL COMPLEXITY ANALYSIS

Let $N$, $D$, $E$ the number of nodes, features and edges of a graph $\mathcal{G}$, the computational cost of the Dirichlet energy is $O(E \times D)$, while the cost of the Projection Energy is $O(N \times D)$. In contrast, most of the computational cost for the numerical rank and the effective rank is given by the computation of the spectral radius of $X$, and of all the singular values of $X$ for the effective rank, respectively.

Standard results show that it is always possible to compute the full singular value decomposition (SVD), and subsequently the spectral radius, in $O(N \times D \times \min\{N, D\})$. However, in typical cases, the latter cost can be drastically reduced using different strategies. Firstly, in the case of numerical rank, the computational cost of the 2-norm is generally much smaller than the cost of the full SVD. Indeed, using Lanczos or power methods to compute it, the cost scales as $O(N \times D)$, and the methods converge typically very fast. In addition, both the effective rank and the numerical rank can be efficiently controlled by computing only the $k$-largest singular values of $X$. In particular, a truncated SVD containing the $k$-largest singular values can be computed in $O(N \times D \times k)$ using either deterministic algorithms or randomized SVD methods.

In general, the computational cost of metrics to quantify oversmoothing is marginal compared to the cost of training. In production, measuring the emergence of oversmoothing is done by training the model and checking the performance on the validation and test sets as the number of layers changes. The cost of additionally computing effective or numerical ranks is marginal. Moreover, we note that computing the metrics on a subsection of a graph can be sufficiently informative, and as a consequence, most oversmoothing metrics studied in this paper can be computed in less than $10\ ms$ for a graph (or a subsection of it) with less than 2000 nodes.

## D ADDITIONAL EXPERIMENTAL RESULTS WITH SYNTHETIC WEIGHTS

In this section, we conduct an asymptotic ablation study using randomly sampled (synthetic) untrained weights. The aim of these experiments is twofold:

- to demonstrate that similar untrained asymptotic experiments are inherently unrealistic, despite being extensively used in the literature (Wang et al., 2022; Rusch et al., 2022; 2023b; Wu et al., 2023; Roth, 2024; Wang et al., 2025), as they fail to reliably capture oversmoothing in shallower GNNs where the performance degradation occurs.
- to examine the convergence properties of different oversmoothing metrics with weight size control and to empirically validate Theorems 5.1 and 5.3.

We construct a 10-node Barabasi-Albert graph with each node having 32 features. The weights are either an identity matrix or randomly sampled at each layer from a uniform distribution $\mathcal{U}(0, s)$, where $s$ depends on the settings: small weights ($s = 0.05$) lead to an exponentially decaying $\|X^{(l)}\|_F$, and large weights ($s = 0.1$) lead to an exploding $\|X^{(l)}\|_F$ for uncapped activation functions. For identity weights, $\|X^{(l)}\|_F$ is roughly constant (LReLU) or slowly decaying (Tanh). The feature initialization $X^{(0)}$ is sampled from $\mathcal{U}(0, 1)$, and is iterated over 300 layers.

In this asymptotic synthetic setting, as presented in Table 2, the normalized $E_{\mathrm{Dir}}$ and $E_{\mathrm{Proj}}$ exhibit decay patterns similar to those of the effective rank and numerical rank, suggesting these metrics are equally sensitive to asymptotic rank collapse. However, this behaviour stands in stark contrast to results on trained networks presented in Tables 1 and 3 and Appendix G, where the normalized $E_{\mathrm{Dir}}$ and $E_{\mathrm{Proj}}$ often fail to detect oversmoothing.

Moreover, these asymptotic results validate Theorems 5.1 and 5.3, showing that the numerical rank converges to one for GCN + LReLU and GAT + any subhomogeneous activation functions. Without making any additional assumption on the normalization of the adjacency matrix, the effective rank and numerical rank do not generally decay to one when subhomogeneous activation functions, e.g. Tanh, are used in GCNs.

| Architecture | $E_{\mathrm{Dir}}$ | | $E_{\mathrm{Proj}}$ | | MAD | Erank | NumRank |
| --- | --- | --- | --- | --- | --- | --- | --- |
| | standard | normalized | standard | normalized | | | |
| GCN+LReLU+identity weights | ✓ | ✓ | ✓ | ✓ | ✓ | ✓ | ✓ |
| GCN+Tanh+identity weights | ✓ | ✓ | ✓ | ✓ | ✗ | ✓ | ✓ |
| GAT+LReLU+identity weights | ✓ | ✓ | ✓ | ✓ | ✓ | ✓ | ✓ |
| GAT+Tanh+identity weights | ✓ | ✓ | ✓ | ✓ | ✗ | ✓ | ✓ |
| GCN+LReLU+small weights | ✓ | ✓ | ✓ | ✓ | ✗ | ✓ | ✓ |
| GCN+Tanh+small weights | ✓ | ✓ | ✓ | ✓ | ✗ | ✓ | ✓ |
| GAT+LReLU+small weights | ✓ | ✓ | ✓ | ✓ | ✗ | ✓ | ✓ |
| GAT+Tanh+small weights | ✓ | ✓ | ✓ | ✓ | ✗ | ✓ | ✓ |
| GCN+LReLU+large weights | ✗ | ✓ | ✗ | ✓ | ✓ | ✓ | ✓ |
| GCN+Tanh+large weights | ✗ | ✗ | ✗ | ✗ | ✗ | ✗ | ✗ |
| GAT+LReLU+large weights | ✗ | ✓ | ✗ | ✓ | ✓ | ✓ | ✓ |
| GAT+Tanh+large weights | ✓ | ✓ | ✓ | ✓ | ✓ | ✓ | ✓ |

Table 2: Additional results on very deep (300 layers) synthetic networks with randomly sampled weights. For Erank and NumRank, we subtract 1 so that both metrics converge to zero. ✓ indicates a decay of the corresponding metric to zero, ✗ indicates otherwise. Note that GAT has similar asymptotic behaviour to GCN with adjacency normalization $D^{-1}\widetilde{A}$.

# E ADDITIONAL EXPERIMENTAL RESULTS ON ACTIVATION FUNCTIONS AND DIFFERENT DATASETS

We extend Table 1 to subhomogenous Tanh activation function and a few additional datasets. The experimental setting is consistent with that of Section 6.

| Dataset | Architecture | $E_{\text{Dir}}$ Standard | $E_{\text{Dir}}$ Normalized | $E_{\text{Proj}}$ Standard | $E_{\text{Proj}}$ Normalized | MAD | Erank | NumRank | Accuracy ratio |
|---|---|---|---|---|---|---|---|---|---|
| Cora | GCN+LReLU | -0.7871 | 0.6644 | -0.8106 | -0.8309 | -0.2460 | **0.9724** | 0.5885 | 0.2693 |
| | GCN+Tanh | 0.5243 | 0.9403 | 0.8610 | 0.9768 | 0.9734 | **0.9923** | 0.9784 | 0.1937 |
| | GAT+LReLU | -0.9189 | 0.6703 | -0.9469 | -0.6054 | 0.8251 | **0.9722** | 0.7612 | 0.2493 |
| | GAT+Tanh | 0.8300 | 0.8501 | 0.8676 | 0.9066 | 0.8603 | **0.9600** | 0.9515 | 0.1900 |
| Citeseer | GCN+LReLU | -0.8442 | 0.4350 | -0.8913 | -0.8667 | -0.7169 | **0.9700** | 0.6795 | 0.4380 |
| | GCN+Tanh | 0.3420 | 0.8957 | 0.4631 | 0.9045 | 0.9605 | **0.9906** | 0.9457 | 0.3509 |
| | GAT+LReLU | -0.9576 | 0.0664 | -0.9585 | -0.9080 | 0.3722 | **0.9915** | 0.8047 | 0.4672 |
| | GAT+Tanh | 0.9234 | 0.8949 | 0.8276 | 0.8997 | 0.9176 | **0.9287** | 0.9024 | 0.3045 |
| Pubmed | GCN+LReLU | -0.9068 | 0.7006 | -0.8508 | -0.1109 | 0.6205 | **0.9464** | 0.9268 | 0.5225 |
| | GCN+Tanh | -0.2330 | 0.2657 | 0.2029 | 0.9137 | 0.8745 | 0.9328 | **0.9940** | 0.3883 |
| | GAT+LReLU | -0.8735 | -0.3684 | -0.8541 | -0.4102 | -0.3932 | 0.9270 | **0.9721** | 0.5564 |
| | GAT+Tanh | 0.2977 | 0.8411 | 0.7160 | **0.9331** | 0.8546 | 0.9303 | 0.8551 | 0.4464 |
| Squirrel | GCN+LReLU | -0.7774 | 0.4171 | -0.7602 | -0.3258 | -0.8247 | 0.6316 | **0.9582** | 0.8457 |
| | GCN+Tanh | 0.6026 | 0.7736 | 0.1689 | 0.9377 | 0.7727 | 0.9680 | **0.9837** | 0.8152 |
| | GAT+LReLU | -0.6864 | -0.5503 | -0.7364 | -0.7253 | 0.5002 | **0.8538** | 0.6840 | 0.7533 |
| | GAT+Tanh | -0.3606 | -0.6557 | -0.0363 | 0.8714 | -0.7033 | **0.8911** | 0.8861 | 0.9103 |
| Chameleon | GCN+LReLU | -0.9223 | 0.1504 | -0.9163 | -0.8201 | -0.8809 | **0.9387** | 0.9014 | 0.6195 |
| | GCN+Tanh | 0.2742 | 0.8796 | -0.2541 | 0.8492 | 0.9272 | **0.9869** | 0.9841 | 0.7093 |
| | GAT+LReLU | -0.8721 | 0.1942 | -0.9089 | -0.8234 | 0.2803 | **0.9446** | 0.8799 | 0.6332 |
| | GAT+Tanh | 0.4090 | 0.5699 | 0.0743 | 0.8230 | 0.6006 | **0.9613** | 0.9143 | 0.7052 |
| Amazon Ratings | GCN+LReLU | -0.9297 | 0.8809 | -0.9079 | -0.3423 | 0.9201 | **0.9301** | 0.8049 | 0.8562 |
| | GCN+Tanh | -0.6960 | -0.6289 | -0.6910 | 0.8576 | 0.9423 | **0.9871** | 0.9327 | 0.8574 |
| | GAT+LReLU | -0.9388 | 0.5277 | -0.9089 | -0.1617 | 0.6545 | **0.9248** | 0.8764 | 0.8384 |
| | GAT+Tanh | 0.8354 | 0.8507 | -0.6595 | 0.9245 | **0.9418** | 0.8954 | 0.8883 | 0.8382 |
| Roman Empire | GCN+LReLU | -0.5635 | **0.7703** | -0.6772 | 0.2575 | 0.6420 | 0.5833 | 0.5368 | 0.3891 |
| | GCN+Tanh | 0.8018 | **0.9225** | -0.6808 | 0.8359 | 0.8124 | 0.8570 | 0.8090 | 0.4067 |
| | GAT+LReLU | -0.9174 | 0.5390 | -0.9407 | 0.1868 | 0.7582 | 0.7221 | **0.8722** | 0.3705 |
| | GAT+Tanh | 0.5767 | 0.5844 | -0.4716 | **0.8819** | 0.7263 | 0.7332 | 0.8589 | 0.3652 |
| OGB-Arxiv | GCN+LReLU | 0.7738 | 0.9194 | 0.5740 | -0.2738 | 0.2822 | **0.9682** | 0.9091 | 0.0957 |
| | GCN+Tanh | -0.6409 | 0.7041 | -0.6808 | 0.9494 | 0.9487 | **0.9900** | 0.9876 | 0.1204 |
| | GAT+LReLU | -0.4097 | 0.9439 | -0.7230 | 0.8985 | 0.8492 | 0.7740 | **0.9781** | 0.2310 |
| | GAT+Tanh | 0.7834 | 0.7896 | -0.9277 | **0.9393** | 0.8222 | 0.7671 | 0.7861 | 0.2376 |
| Average correlation | | -0.1956 | 0.5137 | -0.3886 | 0.2669 | 0.4960 | **0.9007** | 0.8684 | |

Table 3: Additional correlation coefficient results on homophilic (Cora, Citeseer, Pubmed), heterophilic (Squirrel, Chameleon, Amazon Ratings, Roman Empire) and large-scale (OGB-Arxiv) dataset.

# F ADDITIONAL EXPERIMENTAL RESULTS WITH DIFFERENT NETWORK COMPONENTS

Prior literature indicates that when adding additional components, such as bias (Rusch et al., 2023a) or residual terms (Scholkemper et al., 2024), the Dirichlet energy does not decay. Therefore, we compare the correlation coefficients in Table 4 between existing oversmoothing metrics when additional components are added, such as bias, LayerNorm, BatchNorm, PairNorm (Zhao & Akoglu, 2019), DropEdge (Rong et al., 2019; Huang et al., 2020) and residual connections (Scholkemper et al., 2024).

All experiments follow the setup described in Section 6. In addition, DropEdge has a probability of 0.5 in removing each edge at each layer. The residual connection is implemented as follows

$$X^{(l+1)} = \sigma(AX^{(l)}W_1^{(l)}) + X^{(0)}W_2^{(l)}.$$

Table 4 demonstrates that both the effective rank and numerical rank achieve a higher average correlation with the classification accuracy than alternative metrics. This finding confirms their superior consistency in detecting oversmoothing across a variety of architectural variants.

Furthermore, we note that when oversmoothing is effectively alleviated, e.g. when residual connection is used, all metrics have a poor correlation with the classification accuracy. This generic limitation is discussed in Appendix B.

| Architecture | $E_{\mathrm{Dir}}$ | | $E_{\mathrm{Proj}}$ | | MAD | Erank | NumRank | Accuracy ratio |
| --- | --- | --- | --- | --- | --- | --- | --- | --- |
| | Standard | Normalized | Standard | Normalized | | | | |
| GCN+LReLU+Bias | -0.9300 | 0.7505 | -0.9414 | -0.3642 | -0.2828 | **0.9847** | 0.7833 | 0.2110 |
| GCN+Tanh+Bias | 0.4086 | 0.8950 | 0.7479 | 0.9026 | 0.8996 | **0.9926** | 0.9814 | 0.2133 |
| GCN+LReLU+LayerNorm | -0.9132 | 0.8445 | -0.9424 | 0.5591 | **0.9753** | 0.9736 | 0.9736 | 0.4882 |
| GCN+Tanh+LayerNorm | -0.7119 | 0.2560 | -0.2069 | 0.9716 | 0.8695 | 0.9576 | **0.9684** | 0.1886 |
| GCN+LReLU+BatchNorm | 0.8789 | 0.7300 | **0.8872** | 0.7175 | 0.5094 | 0.6005 | 0.6577 | 0.8761 |
| GCN+Tanh+BatchNorm | -0.6033 | 0.2883 | -0.6587 | **0.7717** | 0.4984 | 0.7377 | 0.7038 | 0.8157 |
| GCN+LReLU+PairNorm | -0.8165 | 0.3731 | -0.8106 | 0.3885 | 0.4850 | 0.5597 | **0.6244** | 0.8556 |
| GCN+Tanh+PairNorm | -0.5963 | 0.0346 | -0.5401 | -0.3358 | -0.0631 | **0.9838** | 0.9298 | 0.3123 |
| GCN+LReLU+DropEdge | -0.7497 | 0.7185 | -0.7939 | -0.7974 | -0.4619 | **0.9720** | 0.5782 | 0.2515 |
| GCN+Tanh+DropEdge | 0.2319 | 0.8388 | 0.5872 | 0.8887 | 0.8763 | **0.9934** | 0.9558 | 0.2096 |
| GCN+LReLU+Residual | -0.0857 | -0.3072 | -0.0559 | -0.1611 | -0.1656 | -0.2466 | -0.1296 | 1.0046 |
| GCN+Tanh+Residual | -0.6751 | -0.7406 | -0.3031 | -0.4019 | -0.6043 | -0.5425 | -0.4897 | 1.0122 |
| Average correlation | -0.3801 | 0.3901 | -0.2525 | 0.2616 | 0.2946 | **0.6638** | 0.6280 | |

Table 4: Additional correlation coefficient results on GCNs with different network components.

## G    METRIC BEHAVIOUR EXAMPLES ON CORA, CITESEER AND PUBMED

We extend Figure 2 to additional datasets and network components. The experimental setting is consistent with that of Section 6.

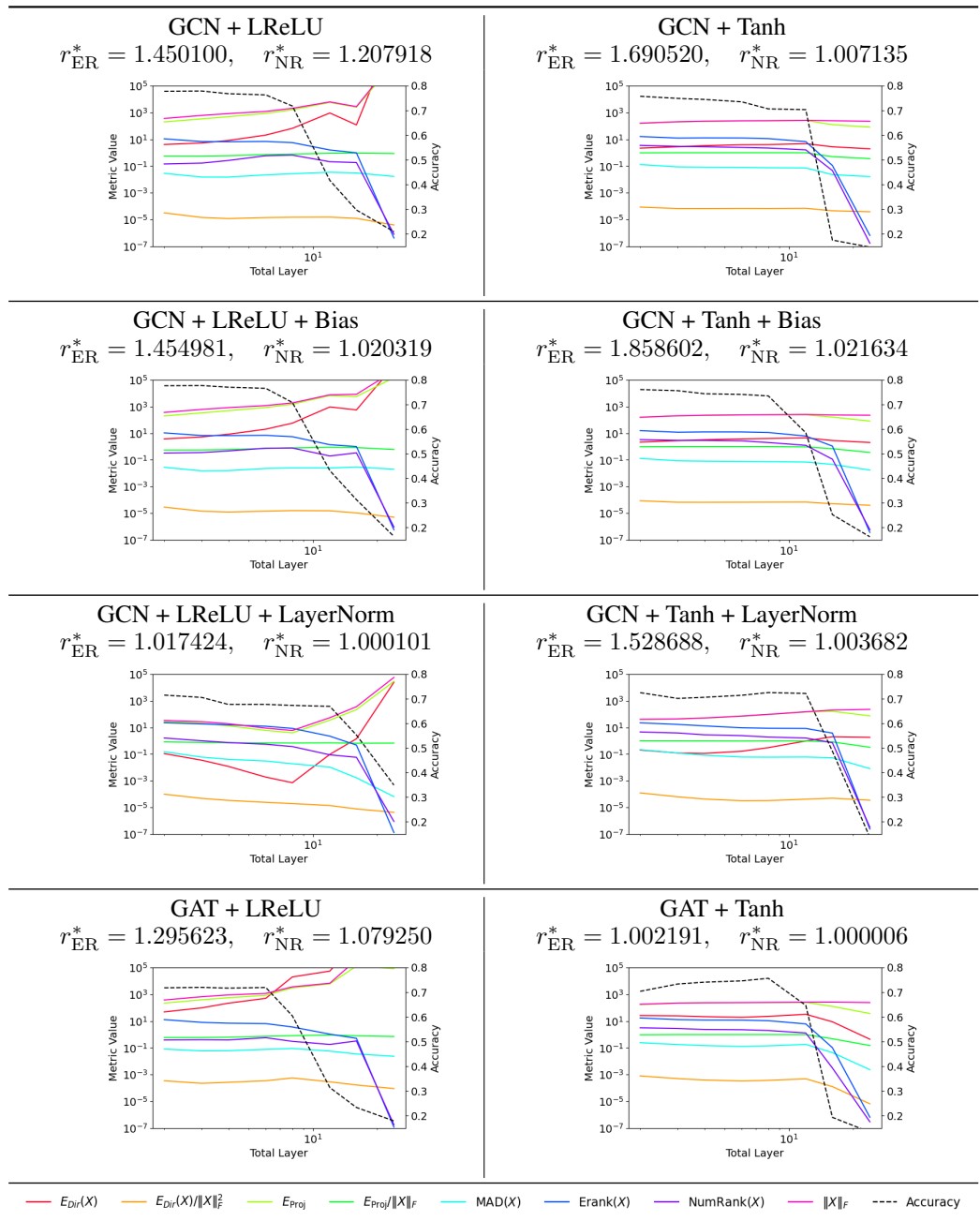

Table 5: The table showcases the behaviour of different metrics and the classification accuracies for 8 GNNs separately trained on Cora Dataset. This table is an extension of figure 2

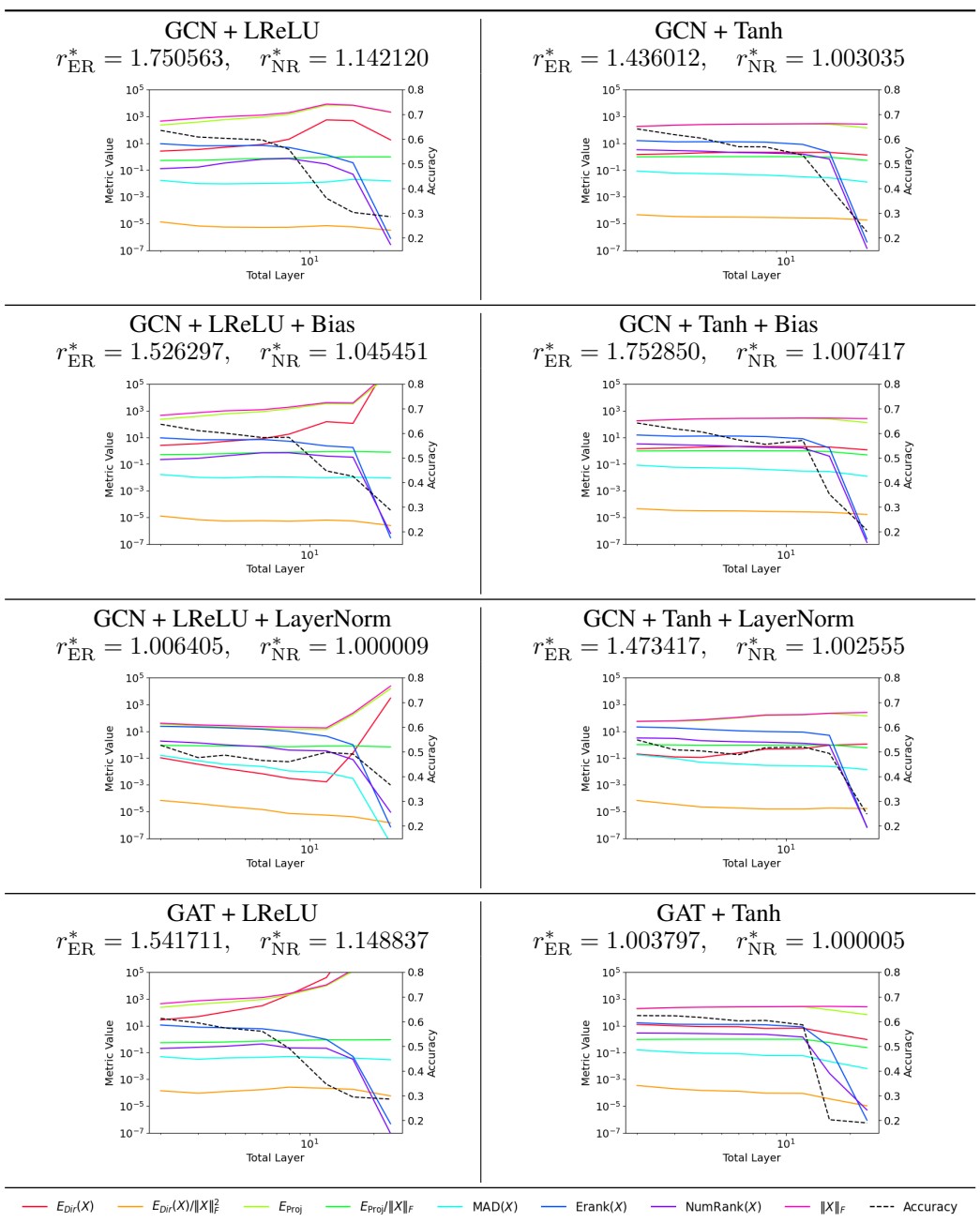

Table 6: The table showcases the behaviour of different metrics and the classification accuracies for 8 GNNs separately trained on Citeseer Dataset.

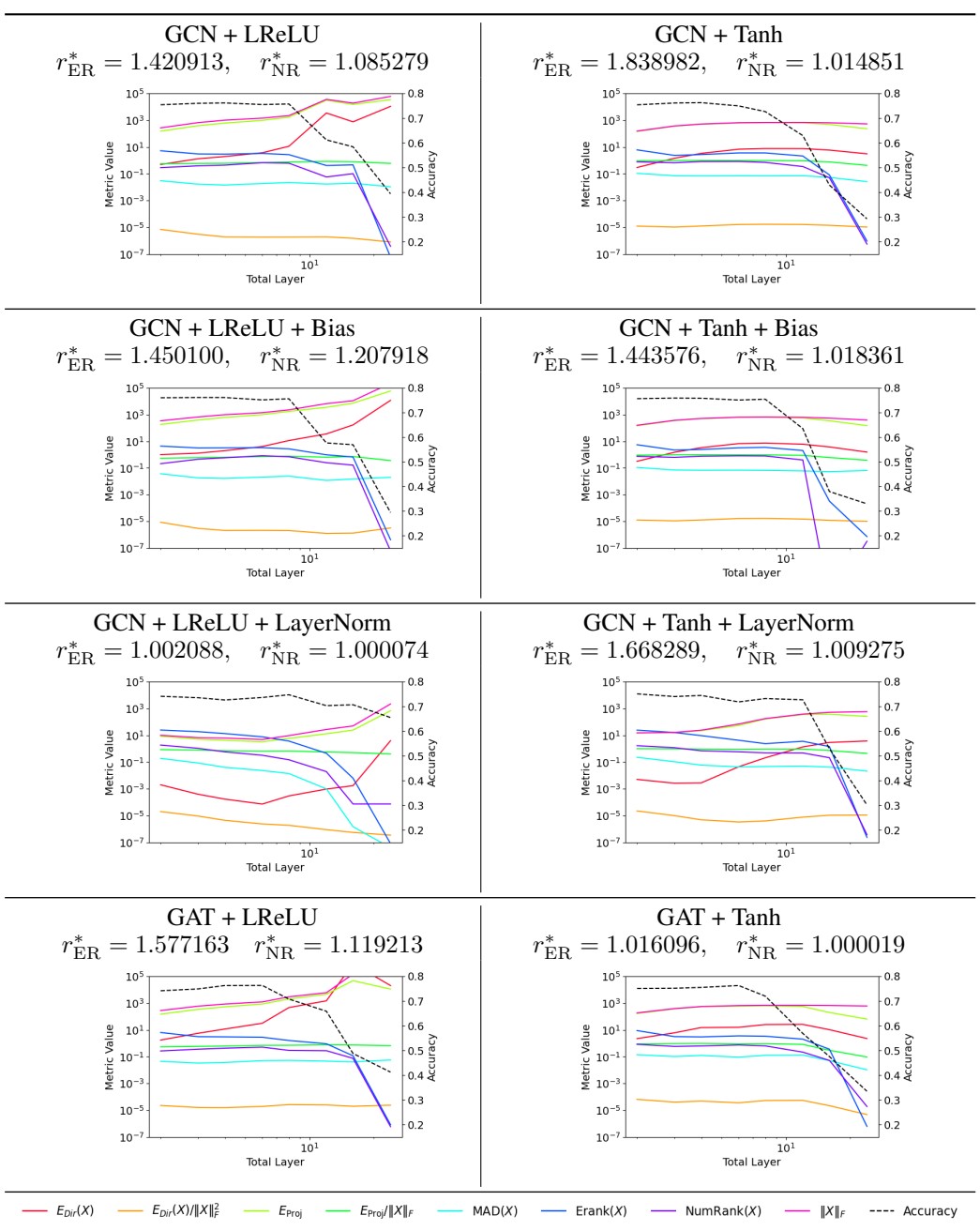

Table 7: The table showcases the behaviour of different metrics and the classification accuracies for 8 GNNs separately trained on Pubmed Dataset.

