# OpenReview forum: "Are we measuring oversmoothing in graph neural networks correctly?"
_ICLR.cc/2026/Conference — ICLR 2026 Poster_

### Official Review · Reviewer_472b · 2025-10-27

**Soundness:** 3
**Presentation:** 3
**Contribution:** 2
**Rating:** 4
**Confidence:** 2

**Summary:**

This paper investigates widely used measures of over-smoothing in graph neural networks such as Dirichlet energy and investigates alternative metrics based on the rank of the feature matrix and argues that rank based measures are better for measuring oversmoothing. The paper reviews existing results on oversmoothign and extends them to rank based measures. Finally, the paper experiments on trained GNNs and shows that the rank based measures correlates better with performance on many datasets.

**Strengths:**

1. **Significance of problem.** Developing improved tools for understanding the internal mechanisms of GNNs is an important and timely direction.

2. **Insightful discussion of limitations.** The paper provides a thorough critique of standard oversmoothing metrics, clearly articulating where existing measures fall short.

3. **Potential of rank-based metrics.** The proposed rank-based metrics offer a promising way to capture more nuanced behavior than conventional oversmoothing measures, potentially enabling a more fine-grained understanding of representation collapse in GNNs.

**Weaknesses:**

1. **Lack of relevance** Over-smoothing as a problem has been widely studied, and remedies have been found, the simplest of which is adding a residual connection. This paper proposes a new method to measure a phenomenon that is no longer a problem in practice which weakens its relevance today. This is further shown in Table 4 where little correlation exists between performance and the proposed measures once residual connections are added.
2. **Lack of novelty** A large portion of the paper discusses already established results rather than focusing on what new insight rank based measures can offer.

**Questions:**

1. What does Section 3.2 offer that prior work does not?
2. For Theorem 5.3 could you maybe expand on “Note that this is implied by the relative metrics [like normalized Dirichlet going to 0], but is actually quite weaker ”. Does the theorem lead to a concrete example where the rank is 1 but the normalized Dirichlet energy or E_proj is not zero? This might help the reader understand in what ways this measure offers a more nuanced perspective.
2. Table1: What is the sample size used to compute each correlation? Can you report confidence intervals (or p-values) for the correlation coefficients?
3. Table 1: In many cases standard measures have a strong negative correlation with performance, sometimes even surpassing some of the rank based rank-based measures in absolute correlation. How do you explain this?

---

> ### Author Response · Authors · 2025-11-21
>
> We would like to thank the reviewer for their recognition of the significance of our work and its insightful comparisons of rank-based metrics with other existing oversmoothing metrics.
>
> We appreciate the reviewer's opinion but respectfully disagree with the statement that over-smoothing lacks relevance. In fact, we feel confused by this statement, as it was mentioned as one of the strengths that "understanding the internal mechanisms of GNNs is an important and timely direction."
>
> We also notice that OSM is observed and studied by very recent papers using known-to-be-flawed metrics such as the Dirichlet energy, and some of them even spotlighted at recent conferences, e.g. [1-4]. This demonstrates a significant awareness gap in the community about the energy-based metrics, and our work specifically aims to bridge this gap.
>
> Even if remedies have been found, that indeed we have considered and discussed in our work, a complete understanding of the causes of oversmoothing and also a correct mathematical quantification of this phenomenon are still missing prior to our work.
>
> We believe these are concrete, relevant, and timely problems. Whenever a new GNN architecture is proposed, it’s crucial that it doesn’t suffer from such pathologies. We hope the reviewer agrees that our work partially addresses this gap by highlighting significant issues with typical choices of measures and proposing a more empirically robust class of rank-based metrics.
>
> We provide below a detailed response to the questions raised. We hope this response satisfactorily clarifies the relevance of our paper, and the reviewer may reconsider their score.
>
>
> [1] Epping, Bastian, et al. "Graph neural networks do not always oversmooth." Advances in Neural Information Processing Systems 37 (2024): 48164-48188.
>
> [2] Wang, Yuanqing, and Kyunghyun Cho. "Non-convolutional graph neural networks." Advances in Neural Information Processing Systems 37 (2024): 32705-32730.
>
> [3] Wang, Keqin, et al. "Resolving Oversmoothing with Opinion Dissensus." arXiv preprint arXiv:2501.19089 (2025).
>
> [4] Zhao, Weichen, et al. "Understanding Oversmoothing in Diffusion-Based GNNs From the Perspective of Operator Semigroup Theory." Proceedings of the 31st ACM SIGKDD Conference on Knowledge Discovery and Data Mining V. 1. (2025).
>
> ### Questions
>
> > What does Section 3.2 offer that prior work does not?
>
> Section 3.2 does not contain the main results of the paper, however it provides a new concise and unifying understanding on the hypotheses that yield convergence to zero of the Dirichlet energy.
>
> > For Theorem 5.3 could you maybe expand on “Note that this is implied by the relative metrics [like normalized Dirichlet going to 0], but is actually quite weaker ”. Does the theorem lead to a concrete example where the rank is 1 but the normalized Dirichlet energy or E_proj is not zero? This might help the reader understand in what ways this measure offers a more nuanced perspective.
>
> The two relative distances $EDir$ and $Eproj$ go to zero if and only if the relative distance between the features and the dominant eigenvector of the aggregation matrices goes to zero.  However in the theorem the vector could be a generic vector different from the dominant eigenvector of the aggregation matrices. Of course one could consider a different vector also in $EDir$ and $EProj$, however to compute them and so measure oversmoothing using them, this vector has to be known in advance. Using rank based metrics, instead, there is no need to know it.
> To further understand in what ways rank-based metrics offer a more nuanced perspective, formula (4) rigorously proves that the Numerical Rank always converges to $1$, at least as quickly as the relative Projection Energy converges to $0$, while all our experiments show that the opposite is not true.
>
> We include the response to the other questions in a following comment.

---

> > ### Author Response · Authors · 2025-11-21
> >
> > > Table1: What is the sample size used to compute each correlation? Can you report confidence intervals (or p-values) for the correlation coefficients?
> >
> > We state in line 462 that we use 8 different numbers of layers * averaged over 10 separate runs = 80 points (separately trained GNNs) to compute each correlation coefficient. The average correlation in e.g. Table 3 (which extends Table 1), is computed over 8 datasets $\times$ 4 setups $\times$ 80 data points each = 2560 separately trained GNNs, in which case the outperformance of the rank-based metrics can be clearly observed.
> >
> > We also follow the reviewer's suggestion by providing the 95% confidence interval of the coefficients below. We would apologize for not including them in our initial submission as we believed they provide very limited information and do not change the conclusion of our experiments.
> >
> > | Metric | GAT+LReLU  |  GAT+Tanh  |  GCN+LReLU  |  GCN+Tanh  |
> > | :--- | :--- | :--- | :--- | :--- |
> > | $EDir(X)$ | -0.9189 [-0.9855, -0.6078] | 0.8301 [0.3021, 0.9683] | -0.7871 [-0.9596, -0.1852] | 0.5244 [-0.2860, 0.8974] |
> > | $EDir(X)/\|X\|_F$ | 0.6704 [-0.0650, 0.9339] | 0.8501 [0.3628, 0.9723] | 0.6644 [-0.0757, 0.9325] | 0.9404 [0.6986, 0.9894] |
> > | $EProj(X)$ | -0.9469 [-0.9906, -0.7281] | 0.8677 [0.4195, 0.9758] | -0.8106 [-0.9644, -0.2471] | 0.8611 [0.3977, 0.9745] |
> > | $EProj(X)/\|X\|_F$ | -0.6054 [-0.9183, 0.1731] | 0.9067 [0.5593, 0.9832] | -0.8309 [-0.9685, -0.3045] | 0.9769 [0.8734, 0.9960] |
> > | MAD$(X)$ | 0.8251 [0.2877, 0.9673] | 0.8603 [0.3953, 0.9743] | -0.2460 [-0.8102, 0.5549] | 0.9734 [0.8558, 0.9953] |
> > | Erank$(X)$ | 0.9722 [0.8496, 0.9951] | 0.9600 [0.7894, 0.9930] | 0.9724 [0.8506, 0.9952] | 0.9923 [0.9566, 0.9987] |
> > | NumRank$(X)$ | 0.7612 [0.1219, 0.9541] | 0.9515 [0.7493, 0.9914] | 0.5885 [-0.1985, 0.9141] | 0.9784 [0.8814, 0.9962] |
> >
> > We can observe that Erank consistently has higher 95% CI lower / upper bounds than all other metrics.
> >
> > > Table 1: In many cases standard measures have a strong negative correlation with performance, sometimes even surpassing some of the rank based rank-based measures in absolute correlation. How do you explain this?
> >
> > We would like to thank the reviewer for this insightful question. We note that the negative correlation between the standard version of $EDir$ and $EProj$ is not scale-invariant, which means they are driven by $\Vert X \rVert_F$. This can be straightforwardly observed in **Figure 2**, where as the number of layers increases, $\Vert X\rVert_F$ (pink) tends to increase, dominating the behaviour of $EDir(X)$ (red) and $EProj(X)$ (light green), while the normalised variants $EDir(X)/\Vert X\rVert_F$ (brown) and $EProj(X)/ \Vert X \rVert_F$ (dark green) fluctuates much less significantly over the layers.
> >
> > To backup our statement, we extend Table 1 to include the correlation coefficients of $\|X\|_F$ below
> >
> > | Dataset | Model | $EDir(X)$ | $EDir(X) / \lVert X \rVert_F$ | $EProj(X)$ | $EProj(X) / \lVert X \rVert_F$ | $\lVert X \rVert_F$ |
> > | :--- | :--- | :--- | :--- | :--- | :--- | :--- |
> > | **Cora** | GCN | -0.7871 | 0.6644 | -0.8106 | -0.8309 | -0.7818 |
> > | | GAT | -0.9189 | 0.6703 | -0.9469 | -0.6054 | -0.9344 |
> > | **Citeseer** | GCN | -0.8442 | 0.4350 | -0.8913 | -0.8667 | -0.8556 |
> > | | GAT | -0.9576 | 0.0664 | -0.9585 | -0.9080 | -0.9456 |
> > | **Pubmed** | GCN | -0.9068 | 0.7006 | -0.8508 | -0.1109 | -0.8817 |
> > | | GAT | -0.8735 | -0.3684 | -0.8541 | -0.4102 | -0.9505 |
> > | **Squirrel** | GCN | -0.7774 | 0.4171 | -0.7602 | -0.3258 | -0.7764 |
> > | | GAT | -0.6864 | -0.5503 | -0.7364 | -0.7253 | -0.7357|
> > | **Chameleon** | GCN | -0.9223 | 0.1504 | -0.9163 | -0.8201 | -0.9181 |
> > | | GAT | -0.8721 | 0.1942 | -0.9089 | -0.8234 | -0.9051 |
> > | **Amazon Ratings**| GCN | -0.9297 | 0.8809 | -0.9079 | -0.3423 | -0.9186 |
> > | | GAT | -0.9388 | 0.5277 | -0.9089 | -0.1617 | -0.9079 |
> > | **OGB-Arxiv** | GCN | 0.7738 | 0.9194 | 0.5740 | -0.2738 | 0.5408 |
> > | | GAT | -0.4097 | 0.9439 | -0.7230 | 0.8985 | -0.7224|
> >
> >
> > We note that in all cases, $EDir(X)$, $EProj(X)$ and $\Vert X \rVert_F$ have similar correlation coefficients, indicating the standard metrics are driven by the growth of $\Vert X\rVert_F$, while the correlation coefficients of $EDir(X)/\Vert X\rVert_F$ and $EProj(X)/\Vert X \rVert_F$ are often quite different. We hope that the reviewer would agree with us that $\Vert X \rVert_F$ is not an informative indicator of oversmoothing, despite it having a large (but negative) absolute correlation. This is because its growth over the layers can be driven by the large norm of the trained weights, and not correlated to the alignment of the features towards its dominant eigenspace, which defines oversmoothing.
> >
> > This observation is central to our argument that an ideal oversmoothing metric should be scale-invariant, as stated in **line 288**, and the rank-based metrics meet this requirement.

---

> ### Comment · Reviewer_472b · 2025-11-24
>
> I thank the authors for their rebuttal. I have a few additional comments:
>
> - **Relevance.** I maintain my earlier point that, while rigorously studying the inner workings of GNNs is timely and important, the problem of oversmoothing (as captured by energy-like criteria) has been extensively studied, with multiple established solutions. I appreciate that rank-based metrics may offer a different perspective, but I am not yet convinced that the proposed measures provide enough novel insight into oversmoothing to meet the relevance threshold for ICLR.
>
> - **Confidence intervals.** Thank you for adding confidence intervals. However, they seem to correspond to a subset of Table 3 in Appendix E, which I understand is only for Cora. Could you provide confidence intervals for Table 1, as originally requested?
>
> - **Strong negative correlation.** Thank you for the clarification. The additional columns indeed highlight the importance of scale invariance when designing an oversmoothing metric.

---

> > ### Author Response · Authors · 2025-11-25
> > **Official Comment by Authors Part 2 / 2**
> >
> > | GCN | Cora | Citeseer | Pubmed | Squirrel | Chameleon | Amazon ratings | Ogb arxiv |
> > |---|---|---|---|---|---|---|---|
> > | $E_{Dir}(X)$ | -0.7871 [-0.9596, -0.1852] | -0.8442 [-0.9711, -0.3443] | -0.9068 [-0.9832, -0.5598] | -0.7775 [-0.9575, -0.1610] | -0.9223 [-0.9861, -0.6216] | -0.9298 [-0.9875, -0.6529] | 0.7738 [0.1520, 0.9568] |
> > | $E_{Dir}(X)/\|X\|_F$ | 0.6644 [-0.0757, 0.9325] | 0.4350 [-0.3889, 0.8723] | 0.7006 [-0.0081, 0.9408] | 0.4171 [-0.4073, 0.8670] | 0.1504 [-0.6200, 0.7732] | 0.8809 [0.4647, 0.9783] | 0.9081 [0.5648, 0.9834] |
> > | $E_{Proj}(X)$ | -0.8106 [-0.9644, -0.2471] | -0.8913 [-0.9803, -0.5019] | -0.8508 [-0.9725, -0.3650] | -0.7602 [-0.9539, -0.1196] | -0.9164 [-0.9850, -0.5977] | -0.9079 [-0.9834, -0.5643] | 0.5740 [-0.2194, 0.9104] |
> > | $E_{Proj}(X)/\|X\|_F$ | -0.8309 [-0.9685, -0.3045] | -0.8667 [-0.9756, -0.4162] | -0.1109 [-0.7565, 0.6441] | -0.3259 [-0.8381, 0.4917] | -0.8202 [-0.9663, -0.2736] | -0.3424 [-0.8435, 0.4775] | -0.2739 [-0.8202, 0.5338] |
> > | MAD$(X)$ | -0.2460 [-0.8102, 0.5549] | -0.7169 [-0.9445, -0.0247] | 0.6205 [-0.1496, 0.9220] | -0.8247 [-0.9673, -0.2866] | -0.8810 [-0.9783, -0.4650] | 0.9201 [0.6128, 0.9857] | 0.2822 [-0.5273, 0.8232] |
> > | ERank$(X)$ | 0.9724 [0.8506, 0.9952] | 0.9700 [0.8383, 0.9947] | 0.9464 [0.7256, 0.9905] | 0.6317 [-0.1315, 0.9247] | 0.9388 [0.6916, 0.9891] | 0.9301 [0.6542, 0.9875] | 0.9682 [0.8295, 0.9944] |
> > | NumRank$(X)$ | 0.5885 [-0.1985, 0.9141] | 0.6795 [-0.0483, 0.9360] | 0.9268 [0.6404, 0.9869] | 0.9582 [0.7808, 0.9926] | 0.9014 [0.5393, 0.9822] | 0.8049 [0.2315, 0.9632] | 0.9092 [0.5691, 0.9836] |
> >
> > | GAT | Cora | Citeseer | Pubmed | Squirrel | Chameleon | Amazon ratings | Ogb arxiv |
> > |---|---|---|---|---|---|---|---|
> > | $E_{Dir}(X)$ | -0.9189 [-0.9855, -0.6078] | -0.9576 [-0.9925, -0.7779] | -0.8735 [-0.9769, -0.4390] | -0.6864 [-0.9376, 0.0354] | -0.8721 [-0.9766, -0.4342] | -0.9388 [-0.9891, -0.6917] | -0.4097 [-0.8647, 0.4147] |
> > | $E_{Dir}(X)/\|X\|_F$ | 0.6704 [-0.0650, 0.9339] | 0.0664 [-0.6696, 0.7366] | -0.3684 [-0.8519, 0.4542] | -0.5503 [-0.9043, 0.2522] | 0.1942 [-0.5914, 0.7907] | 0.5277 [-0.2818, 0.8983] | 0.9439 [0.7144, 0.9901] |
> > | $E_{Proj}(X)$ | -0.9469 [-0.9906, -0.7281] | -0.9585 [-0.9927, -0.7821] | -0.8541 [-0.9731, -0.3754] | -0.7364 [-0.9488, -0.0660] | -0.9089 [-0.9836, -0.5680] | -0.9089 [-0.9836, -0.5681] | -0.7230 [-0.9458, -0.0374] |
> > | $E_{Proj}(X)/\|X\|_F$ | -0.6054 [-0.9183, 0.1731] | -0.9080 [-0.9834, -0.5644] | -0.4102 [-0.8649, 0.4143] | -0.7253 [-0.9463, -0.0421] | -0.8234 [-0.9670, -0.2828] | -0.1617 [-0.7777, 0.6128] | 0.8985 [0.5285, 0.9817] |
> > | MAD$(X)$ | 0.8251 [0.2877, 0.9673] | 0.3722 [-0.4506, 0.8531] | -0.3932 [-0.8597, 0.4308] | 0.5002 [-0.3158, 0.8909] | 0.2803 [-0.5288, 0.8225] | 0.6545 [-0.0932, 0.9302] | 0.8492 [0.3599, 0.9721] |
> > | ERank$(X)$ | 0.9722 [0.8496, 0.9951] | 0.9915 [0.9522, 0.9985] | 0.9270 [0.6410, 0.9870] | 0.8538 [0.3744, 0.9730] | 0.9446 [0.7173, 0.9902] | 0.9248 [0.6318, 0.9865] | 0.7740 [0.1525, 0.9568] |
> > | NumRank$(X)$ | 0.7612 [0.1219, 0.9541] | 0.8047 [0.2311, 0.9632] | 0.9721 [0.8489, 0.9951] | 0.6840 [-0.0399, 0.9370] | 0.8799 [0.4611, 0.9781] | 0.8764 [0.4490, 0.9774] | 0.9781 [0.8796, 0.9962] |
> >
> > Again, we observe that most of the CI lower / upper bounds of the rank-based metrics are higher than all other metrics.
> >
> >
> > We hope these clarifications fully address the reviewer’s concerns, and we are happy to provide any further details if needed.
> >
> > ---
> >
> >
> > [1] Epping, Bastian, et al. "Graph neural networks do not always oversmooth." Advances in Neural Information Processing Systems 37 (2024): 48164-48188.
> >
> > [2] Cai, Chen, and Yusu Wang. "A note on over-smoothing for graph neural networks." arXiv preprint arXiv:2006.13318 (2020).
> >
> > [3] Wu, Xinyi, et al. "Demystifying oversmoothing in attention-based graph neural networks." Advances in Neural Information Processing Systems 36 (2023): 35084-35106.
> >
> > [4] Rusch, T. Konstantin, Michael M. Bronstein, and Siddhartha Mishra. "A survey on oversmoothing in graph neural networks." arXiv preprint arXiv:2303.10993 (2023).
> >
> > [5] Scholkemper, Michael, et al. "Residual connections and normalization can provably prevent oversmoothing in gnns." International Conference on Learning Representations (2024).
> >
> > [6] Arnaiz-Rodriguez, Adrian, and Federico Errica. "Oversmoothing," Oversquashing", Heterophily, Long-Range, and more: Demystifying Common Beliefs in Graph Machine Learning." arXiv preprint arXiv:2505.15547 (2025).

---

> ### Author Response · Authors · 2025-11-25
> **Official Comment by Authors Part 1 / 2**
>
> We would like to thank the reviewer for their response and for acknowledging the importance of scale invariance when designing oversmoothing metrics. We address the additional comments below.
>
> > if a new architecture is shown not to suffer from oversmoothing in the Dirichlet sense, it would also not suffer in the rank sense.
>
>
> We acknowledge that some conceptual similarities exist between Dirchlet-like and  rank-based metrics. In fact, we prove in Formula (4) that the Numerical Rank converges to $1$ at least as quickly as the relative Dirichlet Energy converges to $0$. This implies that if an architecture is shown to suffer OSM in the Dirichlet sense, it must also suffer OSM in the rank sense. **However, we respectfully clarify that the other way round must not be true and this a key point in our paper.** Rank-based measures can detect oversmoothing phenomena that Dirichlet-like metrics miss. We provide several concrete examples below, drawn directly from the paper.
>
> 1. **Empirical counterexample (Appendix F).**  In Appendix F, where we explore the performance of different oversmoothing metrics on different architectures, Erank or NumRank's average correlation with model performance double those of $EDir$ and $EProj$. For example, for **GCN+Tanh+PairNorm**, $EDir$ and $EProj$'s correlation coefficients are close to 0 or ngative, whereas those of Erank and NumRank are close to 1, and the model accuracy drops by a factor of 3 from 2 to 24 layers.  We hope the reviewer would agree this clearly shows that it is not true that if a new architecture is shown not to suffer from oversmoothing in the Dirichlet sense, it would also not suffer in the rank sense.
>
>
> 2. **Scale dependence of Dirichlet metrics in existing literature.** Prior works often (sometimes unintentionally) exploit the scale dependence of Dirichlet-like metrics [1–4]. For example, [1] (NeurIPS 2024) concludes:
>    > “if the initial weights of the network have sufficiently large variance, GCNs do not oversmooth”
>
>    because oversmoothing is measured using unnormalised Dirichlet energy. This claim conflicts with our theoretical and empirical findings. Across our experiments on untrained settings (Appendix D, where "large weights" correspond to weights with large variance, as they are sampled with mean 0) and trained models (e.g. Table 1 and Figure 2), we consistently observe oversmoothing even when weight variance is large. We also tried explicitly increasing weight initialisation variance and observed that it does **not** prevent oversmoothing and can even degrade performance.
>
>
> 3. **Practical issues with correctly computing Dirichlet energy.**  Dirichlet-like energy metrics require knowing the dominant eigenspace ($u_i$ and $u_j$ in Formula 1) in advance. This introduces additional complexity for implementation and it is often neglected in the literature, e.g. [4-6] (A possible failure case: when the dominant eigenvector is uniform, i.e. $u_i=u_j$, $EDir$ will not decay) It is also unclear how one can compute $EDir$ correctly when $u_i, u_j$ are training dependent (e.g. when bias is added).
>
>
>
> > Could you provide confidence intervals for Table 1
>
> We sincerely apologise for the confusion on our side. The previously provided confidence intervals corresponded to a subset of Table 3 (Cora only).
> Below we provide the confidence intervals corresponding to **Table 1**, as originally requested.

---

### Official Review · Reviewer_eo32 · 2025-10-31

**Soundness:** 3
**Presentation:** 3
**Contribution:** 2
**Rating:** 4
**Confidence:** 3

**Summary:**

The paper argues that existing energy-based oversmoothing metrics for GNNs (Dirichlet energy and a projection variant) are unreliable in trained networks, and proposes rank-based measures (numerical rank and effective rank) as better metrics. The authors present (i) a unifying framework based on eigenvectors of nonlinear activations, (ii) a linear analysis predicting dominance of the leading eigenspace, and (iii) a nonlinear analysis using the Hilbert projective metric that yields rank collapse under nonnegativity and subhomogeneity conditions. The authors conduct experiments on GCNs and GATs across multiple datasets and small depths demonstrate that rank-based measures correlate well with model accuracy, whereas energy-based metrics often remain insensitive to oversmoothing in some realistic training settings.

**Strengths:**

The paper is overall clearly written. It clearly points out the failure modes of energy metrics. For example, it shows these metrics are informative mainly in the limit, can be confounded by scaling, and rely on a fixed dominant eigenspace, which often violated by trained GNNs. It also provides a unifying theoretical perspective for previous results on the oversmoothing phenomena measured by Dirichlet-like energies. The nonlinear case with Hilbert-metric argument are technically interesting.

**Weaknesses:**

Despite strong empirical results, the paper lacks a theoretical justification that rank‑based metrics dominate Dirichlet‑style energy measures as oversmoothing indicators in trained GNNs. The analysis shows when energies vanish and that ranks can collapse, but offers no formal comparison, dominance result, or sensitivity guarantee for rank.

**Questions:**

Could the authors explain why effective/numerical rank aligns with accuracy more faithfully than Dirichlet‑style energies in trained networks? In the linear setting, at first glance, Thm. 4.1 and Thm. 5.1 suggest that both energy‑like deviations and rank collapse shrink comparably. How do you reconcile this with the empirical gap you observe? Could you provide intuition or formal statement showing a sensitivity advantage for rank metrics?

---

> ### Author Response · Authors · 2025-11-21
>
> We would like to thank the reviewer for their appreciation of the clarity of the paper and the recognition of the failure mode of the existing energy metrics.
>
> We acknowledge the lack of a theoretical proof that rank‑based metrics dominate Dirichlet‑style energy measures as oversmoothing indicators in trained GNNs. We agree this is desirable, but it is also far from trivial to prove formally.
> Note that both Thm 4.1. and Thm 5.1 prove only an upper bound for the shrinking velocity of the numerical rank of the features. Differently, formula (4) rigorously proves that the Numerical Rank always converges to $1$, at least as quickly as the relative Dirichlet Energy converges to $0$; however, the opposite may not be true, which is what we observe in all our experiments.
>
> We do also provide an intuition of the reason the effective/numeric rank aligns to accuracy more closely than the Dirichlet-like energies in trained networks. As we discuss at the beginning of section 4, an important limitation of absolute Dirichlet metrics is that they indicate oversmoothing only in the limit of infinitely many layers, when their values converge exactly to zero. Since they measure a form of absolute distance, a small but nonzero value does not provide any meaningful information. On the other hand, convergence to zero corresponds to either perfect alignment with the dominant eigenspace or the collapse of the feature representation matrix to the all-zero matrix. While the former is a symptom of oversmoothing, the latter does not necessarily imply oversmoothing. Moreover, this convergence property requires the weights to be strongly bounded. However, in most practical cases, performance degradation is observed even in relatively shallow networks, far from being infinitely deep. The rank instead allows for the detection of oversmoothing in shallow networks without requiring exact convergence to rank one. A small value of the effective rank directly implies that the feature representations are low-rank, suggesting a potentially suboptimal network architecture.
>
> We hope that our comments clarify the reviwer's doubts and they may reconsider their score.

---

### Official Review · Reviewer_BZvC · 2025-11-01

**Soundness:** 3
**Presentation:** 3
**Contribution:** 2
**Rating:** 4
**Confidence:** 2

**Summary:**

This paper critiques traditional oversmoothing metrics in graph neural networks (GNNs), such as Dirichlet energy, for their limitations in detecting performance degradation in realistic, trained models. The authors argue these metrics only work asymptotically in deep, untrained networks and propose rank-based alternatives (numerical and effective rank) that measure feature matrix collapse to low rank.

**Strengths:**

The paper's novel critique of widely used metrics is timely and well-supported, exposing flaws like scale-dependence and eigenspace reliance with examples and simplified theorems. Theoretical contributions, including proofs of rank convergence independent of feature magnitude, extend to GCNs/GATs and offer a unifying eigenvector perspective, and some nonlinear analysis. Empirically, the extensive evaluation across homophilic/heterophilic datasets, depths, activations, and ablations robustly demonstrates rank metrics' superiority, with high correlations and methodological rigor (e.g., log-transformed Pearson analysis). This could practically guide deeper GNN designs.

**Weaknesses:**

The theoretical analysis is restricted to linear/nonnegative models or shared eigenvectors, limiting generalizability to signed graphs or complex activations. Experiments focus solely on node classification, neglecting graph/edge-level tasks or larger/dynamic graphs, and don't prove causation between rank decay and performance. While rank metrics excel in trained settings, the paper somewhat overlooks scenarios where energy metrics succeed (e.g., untrained asymptotics) and lacks discussion of computational costs or hybrid approaches.

**Questions:**

None

---

> ### Author Response · Authors · 2025-11-21
>
> We would like to first thank the reviewer for their acknowledgement of the timely significance of our work, the existing flaws related to the widely used metrics, the strong empirical performance, and the extensiveness of our experiments.
>
> We would like to take their attention on the fact that the paper does indeed include in appendix D experiments on untrained GNNs. Results on the larger dataset (OGB-Arxiv) have also been included in Table 1 and 3. We would like to kindly note that the scope of our experiments are in line with the experiments commonly considered in the oversmoothing literature. We will follow the reviewer's suggestion by adding additional experiments below on both graph-level (MUTAG, ENZYMES, PROTEIN) and edge-level (Cora(E), Citeseer(E), Pubmed(E), adapted for edge-prediction task via torch_geometric.transforms.RandomLinkSplit) experiments.
>
> | Dataset | Model | EDir Std | EDir Norm | EProj Std | EProj Norm | MAD | Erank | NumRank |
> | :--- | :--- | :--- | :--- | :--- | :--- | :--- | :--- | :--- |
> | **MUTAG** | GCN | -0.5099 | 0.7477 | -0.5605 | **0.7693** | 0.6029 | 0.6003 | 0.6311 |
> | | GAT | -0.5783 | 0.7882 | -0.5652 | 0.6538 | 0.4832 | **0.8678** | 0.8183 |
> | **ENZYMES** | GCN | -0.8314 | 0.8296 | -0.8419 | 0.7546 | 0.6041 | **0.9202** | 0.8785 |
> | | GAT | -0.8619 | 0.9013 | -0.8424 | 0.9095 | 0.5280 | 0.9447 | **0.9773** |
> | **PROTEINS** | GCN | -0.9868 | **0.9449** | -0.9888 | 0.8487 | 0.8009 | 0.9297 | 0.8547 |
> | | GAT | -0.8600 | 0.8851 | -0.8534 | 0.8887 | 0.9108 | **0.9254** | 0.8818 |
> | **Cora (E)** | GCN | 0.7707 | 0.9379 | 0.7841 | 0.9753 | -0.3587 | **0.9910** | 0.9821 |
> | | GAT | 0.9050 | 0.9672 | 0.9085 | 0.9237 | -0.2713 | 0.9636 | **0.9864** |
> | **Citeseer (E)**| GCN | 0.8582 | 0.9570 | 0.8266 | 0.9514 | -0.5615 | **0.9678** | 0.9673 |
> | | GAT | 0.8760 | 0.9324 | 0.8923 | 0.9340 | -0.7024 | 0.9811 | **0.9961** |
> | **Pubmed (E)** | GCN | 0.7552 | 0.9052 | 0.7473 | **0.9701** | -0.7309 | 0.9428 | 0.9081 |
> | | GAT | 0.8706 | 0.9690 | 0.8235 | 0.9259 | -0.8302 | 0.8435 | **0.9778** |
>
>
>
> Regarding the causation between rank decay and performance drop, note that if the rank of the learned features is smaller than the rank of the output, the learned output resides in a linear subspace with a dimension smaller than the output space. This clearly results in poor approximation properties. For instance, in classification tasks, the output rank will be precisely equal to the number of classes. If the features’ rank falls below that, the performance will inevitably decline. In our experiments, we observe a rank close to 1.
>
> Thank you for pointing out this lack of clarification. We assumed this was somewhat straightforward, but we appreciate that it might not be. We will include a clarifying paragraph in the revised version.
>
>
> About computational costs: We highlight that they are extensively discussed in Appendix C. Our discussion shows that, using appropriate strategies, the computational complexity of ERank and NumRank are comparable (although larger) with the computational cost of the Dirichlet and Projection energy. Nontheless we underline that, since our focus is showing that Dirichlet-like metrics are unreliable to measure oversmoothing, we do not consider a comparison between the computational costs of primary importance. The metrics are mainly used on a research level, and are mostly irrelevant for compute- or memory- sensitive large-scale deployment of any trained model. We would like to also include a small ablation study on the runtime and memory usage of different metrics with a graph of around 1000 nodes and 4000 edges.
> | Metric | DE | E_proj | MAD | Erank | NumRank |
> | :--- | :--- | :--- | :--- | :--- | :--- |
> | **Runtime / ms** | 0.119 ± 0.003 | 0.106 ± 0.007 | 652 ± 30 | 1.69 ± 0.07 | 1.51 ± 0.07 |
> | **Peak memory / MB** | 8.45 | 12.87 | 8.33 | 1.17 | 1.20 |
>
> This demonstrates that the computational costs of most metrics remain at the micro-second level and thus rather negligible.
>
> We are unsure what the reviewer mean with hybrid approach. If it means a combination of Dirichlet energy and rank-based measure, we don't expect it to yield any benefit.
>
>
> Finally, about our theoretical analysis, we agree that it requires a number of assumptions, however we also want to take the reviewer attention on the fact that our theoretical analysis is more general than all the results we are aware of, that it rigorously proves the emergence of overmoothing, and also brings a new perspective on the oversmoothing phenomenon from the point of view of eigenvectors of activation functions.
>
> We hope our response has satisfactorily addressed all the criticisms, and that it can convince the reviewer to reconsider their score upwards.

---

### Official Review · Reviewer_EXwb · 2025-11-01

**Soundness:** 3
**Presentation:** 3
**Contribution:** 2
**Rating:** 4
**Confidence:** 4

**Summary:**

This paper argues that dirichlet energy based measures may fail to capture oversmoothing in real application scenarios, so the author introduces rank (numerical or effective rank) as an effective alternative measurement of oversmoothing. The main contributions of the paper are: 1. The authors provide novel rank relaxation functions to measure oversmoothing. 2. The authors compare current oversmoothing metrics with their relaxed rank metrics and demonstrate rank metrics correlate better with performance, 3. The authors theoretically justify the aggregation matrices and nonlinear activation functions will both contribute to the decay of rank, whereas energy-based measures fail to detect oversmoothing. Empirical results shows that the performance drop from GNN is well-correlated with the decrease of rank metric based on comparisons with different energy and rank metrics.

**Strengths:**

1. This paper provides a strong theoretical proof of the decay of the rank as a measurement of oversmoothing.
2. Extensive experiments and ablation studies show the rank-based metrics correlate better with performance degradation than energy-based measurement with extension to modern architectures with dropout, residuals, or normalization layers.
3. Unlike Dirichlet energy, which only shows meaningful trends for very deep networks, the proposed rank metrics can detect degradation even in moderately deep networks (10 or fewer layers).

**Weaknesses:**

1. Research of related work part is not sufficient as many previous works have already shown energy-based metric is not a proper or sufficient measurement of oversmoothing. For example, [1] has shown that oversmoothing can be mitigated without explicitly dirichlet energy based control.
2. The paper shows rank metrics are effective measurement of oversmoothing but does not provide practical techniques to mitigate it using the rank insight.


[1] Y. Jin and X. Zhu, "Graph Rhythm Network: Beyond Energy Modeling for Deep Graph Neural Networks," 2024 IEEE International Conference on Data Mining (ICDM), Abu Dhabi, United Arab Emirates, 2024, pp. 723-728, doi: 10.1109/ICDM59182.2024.00083.

**Questions:**

What are certain techniques that can mitigate oversmoothing from the view of rank since the authors show ranks metrics are more effective than energy-based metrics. Also, could the rank metric be leveraged as a regulation metric to prevent oversmoothing, or to achieve better performance?

---

> ### Author Response · Authors · 2025-11-21
>
> We would like to first thank the reviewer for their recognition of the strong theoretical support for the proposed metric, the extensiveness of experiments, and the limitation of the existing energy-like metrics.
>
> We also thank the reviewer for drawing our attention to a line of work we were previously unaware of, including [1] and the references therein. Of course ideas, from [1] and [2] could be leveraged to design architectures that avoid or alleviate oversmoothing, e.g. using residual architectures where the magnitude of the residual is regulated according to the rank of the features of neighbooring nodes. We will include a discussion about these possible line of work in the conclusions of the revised version.
>
> However, we would also like to point out that OSM is observed and studied by very recent papers using known-to-be-flawed energy-based metrics, and some of them even spotlighted at recent conferences, e.g. [3-6]. This demonstrates a significant awareness gap in the community about the energy-based metrics, and our work specifically aims to bridge this gap.
>
> Nontheless, we hope the reviewer will agree with us that this goes beyond the current scope of the manuscript, which is understanding rigorously the causes of oversmoothing and how to reliably measure it from a mathematical point of view.
>
> In light of our response and of the overall positive feedback, as also the fact that our work could open possibly new future lines of work, we hope the reviewer may reconsider their score upwards, for the strong empirical performance of the proposed rank-based metrics over the widely used energy-based ones, as well as our strong theoretical support. We remain available for further discussion.
>
>
> [1] Y. Jin and X. Zhu, "Graph Rhythm Network: Beyond Energy Modeling for Deep Graph Neural Networks," in IEEE, 2024.
>
> [2] T. K. Rusch, B. P. Chamberlain, M. W. Mahoney, et al., “Gradient gating
> for deep multi-rate learning on graphs,” in ICLR, 2023.
>
> [3] Epping, Bastian, et al. "Graph neural networks do not always oversmooth." Advances in Neural Information Processing Systems 37 (2024): 48164-48188.
>
> [4] Wang, Yuanqing, and Kyunghyun Cho. "Non-convolutional graph neural networks." Advances in Neural Information Processing Systems 37 (2024): 32705-32730.
>
> [5] Wang, Keqin, et al. "Resolving Oversmoothing with Opinion Dissensus." arXiv preprint arXiv:2501.19089 (2025).
>
> [6] Zhao, Weichen, et al. "Understanding Oversmoothing in Diffusion-Based GNNs From the Perspective of Operator Semigroup Theory." Proceedings of the 31st ACM SIGKDD Conference on Knowledge Discovery and Data Mining V. 1. (2025).

---

### Meta-Review · Area_Chair_FWBa · 2025-12-30

**Summary:**

The authors study the vexing problem of Oversmoothing (OSM) in Graph neural networks. Their principal argument that the currently used Dirichlet energy based metrics of OSM are not suitable for the practical case of trained GNN with 10+ layers and only are useful for very deep networks that are rarely encountered in practice. Their key contribution is a new rank-based metric and they theoretically and empirically demonstrate that their rank-based metric is clearly more sensitive in capturing over smoothing than the energy based ones. They provide extensive empirical demonstration of their approach. Given the strong contributions to a difficult problem, the paper clearly should be accepted.

**Reviewer Concerns:**

The authors have satisfactorily addressed the technical and empirical concerns of the reviewers.

One of the reviewers insisted on providing mitigations to over smoothing. While this is a key limitation of the paper, the authors did not claim that their scope pertained to actual mitigation. Rather, they provide a very sensitive metric which can be a target of OSM mitigation in future work.

Another reviewer questioned the relevance of research in over smoothing. The authors have correctly pointed out that it remains an active area of research in the GNN community even now.

**Reviewer Scores:**

Reviewer  EXwb scored this paper as a Borderline reject with the main concern that the paper does not address OSM mitigation. The authors have correctly pointed out that it is beyond their current scope. Most likely, this would have encouraged the reviewer to raise the score to borderline accept, if not more.

Reviewer BZvC  scored this paper as a Borderline reject with the main concern of how the new metric relates to existing Dirichlet energy type metrics. The authors have correctly pointed out that their metric dominates the energy metric in the sense that it will always be more sensitive than the energy metric. This should be sufficient in assuaging the concern of the reviewer and raising the score to borderline accept, if not more.


Reviewer eo32 scored this paper as a Borderline reject with the main concern of the need of an explicit dominance relation between the rank and Dirichlet energy metrics. The authors correctly pointed out that they do so in some sense and an even more precise bound in the non-asymptotic regime is infeasible as the Dirichlet energy itself is an asymptotic indicator. This should be sufficient in assuaging the concern of the reviewer and raising the score to borderline accept, if not more.

Reviewer  472b also scored this paper as a Borderline reject questioning the relevance of research on OSM in the current context. The authors have carefully provided a list of recent references on OSM in leading ML venues. This might not be sufficient to raise the score as the reviewer questions the very rationale of the paper.

On balance, the AC is convinced that the authors study a relevant problem and provide major contributions, leading to a recommendation of acceptance.

---

### Decision · Program_Chairs · 2026-01-26

Accept (Poster)